# Optimistic Mirror Descent in Saddle-Point Problems: Going the Extra (Gradient) Mile

**Panayotis Mertikopoulos**
Univ. Grenoble Alpes, CNRS, Inria, Grenoble INP, LIG
38000 Grenoble, France
`panayotis.mertikopoulos@imag.fr`

**Bruno Lecouat, Houssam Zenati, Chuan-Sheng Foo, Vijay Chandrasekhar**
Institute for Infocomm Research, A*STAR
1 Fusionopolis Way, #21-01 Connexis (South Tower), Singapore
`{bruno_lecouat,houssam_zenati,foocs,vijay}@i2r.a-star.edu.sg`

**Georgios Piliouras**
Singapore University of Technology and Design
8 Somapah Road, Singapore
`georgios@sutd.edu.sg`

## Abstract

Owing to their connection with generative adversarial networks (GANs), saddle-point problems have recently attracted considerable interest in machine learning and beyond. By necessity, most theoretical guarantees revolve around convex-concave (or even linear) problems; however, making theoretical inroads towards efficient GAN training depends crucially on moving beyond this classic framework. To make piecemeal progress along these lines, we analyze the behavior of mirror descent (MD) in a class of non-monotone problems whose solutions coincide with those of a naturally associated variational inequality – a property which we call *coherence*. We first show that ordinary, "vanilla" MD converges under a strict version of this condition, but not otherwise; in particular, it may fail to converge even in bilinear models with a unique solution. We then show that this deficiency is mitigated by optimism: by taking an "extra-gradient" step, optimistic mirror descent (OMD) converges in *all* coherent problems. Our analysis generalizes and extends the results of Daskalakis et al. [2018] for optimistic gradient descent (OGD) in bilinear problems, and makes concrete headway for provable convergence beyond convex-concave games. We also provide stochastic analogues of these results, and we validate our analysis by numerical experiments in a wide array of GAN models (including Gaussian mixture models, and the CelebA and CIFAR-10 datasets).

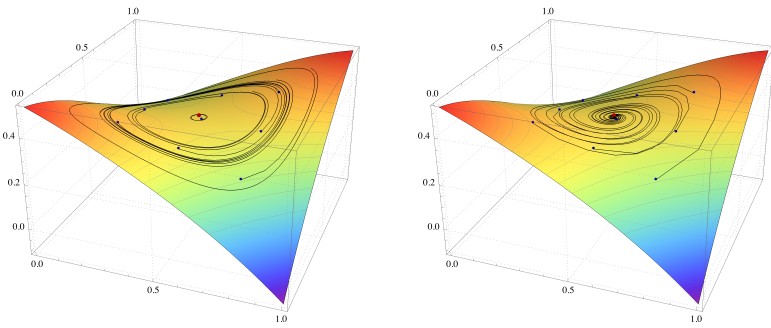

**Figure 1:** Mirror descent (MD) in the non-monotone saddle-point problem $f(x_1, x_2) = (x_1 - 1/2)(x_2 - 1/2) + \frac{1}{3}\exp(-(x_1 - 1/4)^2 - (x_2 - 3/4)^2)$. Left: vanilla MD spirals outwards; right: optimistic MD converges.

# 1 INTRODUCTION

The surge of recent breakthroughs in deep learning has sparked significant interest in solving optimization problems that are universally considered hard. Accordingly, the need for an effective theory has two different sides: first, a deeper understanding would help demystify the reasons behind the success and/or failures of different training algorithms; second, theoretical advances can inspire effective algorithmic tweaks leading to concrete performance gains. For instance, using tools from the theory of dynamical systems, Lee et al. [2016, 2019] and Panageas & Piliouras [2017] showed that a wide variety of first-order methods (including gradient descent and mirror descent) almost always avoid saddle points. More generally, the optimization and machine learning communities alike have dedicated significant effort in understanding non-convex landscapes by searching for properties which could be leveraged for efficient training. As an example, the "strict saddle" property was shown to hold in a wide range of salient objective functions ranging from low-rank matrix factorization [Bhojanapalli et al., 2016; Ge et al., 2017] and dictionary learning [Sun et al., 2017a,b], to principal component analysis [Ge et al., 2015], and many other models.

On the other hand, *adversarial* deep learning is nowhere near as well understood, especially in the case of generative adversarial networks (GANs) [Goodfellow et al., 2014]. Despite an immense amount of recent scrutiny, our theoretical understanding cannot boast similar breakthroughs as in "single-agent" deep learning. Because of this, a considerable corpus of work has been devoted to exploring and enhancing the stability of GANs, including techniques as diverse as the use of Wasserstein metrics [Arjovsky et al., 2017], critic gradient penalties [Gulrajani et al., 2017], feature matching, minibatch discrimination, etc. [Radford et al., 2016; Salimans et al., 2016].

Even before the advent of GANs, work on adaptive dynamics in general bilinear zero-sum games (e.g. Rock-Paper-Scissors) established that they lead to persistent, chaotic, recurrent (i.e. cycle-like) behavior [Sato et al., 2002; Piliouras & Shamma, 2014; Piliouras et al., 2014]. Recently, simple specific instances of cycle-like behavior in bilinear games have been revisited mainly through the lens of GANs [Mertikopoulos et al., 2018; Daskalakis et al., 2018; Mescheder et al., 2018; Papadimitriou & Piliouras, 2018]. Two important recent results have established unified pictures about the behavior of continuous and discrete-time first order methods in bilinear games: First, Mertikopoulos et al. [2018] established that continuous-time descent methods in zero-sum games (e.g., gradient descent, follow-the-regularized-leader and the like) are Poincaré recurrent, returning arbitrarily closely to their initial conditions infinitely many times. Second, Bailey & Piliouras [2018] examined the discrete-time analogues (gradient descent, multiplicative weights and follow-the-regularized-leader) showing that orbits spiral slowly outwards. These recurrent systems have formal connections to Hamiltonian dynamics and do not behave in a gradient-like fashion Balduzzi et al. [2018]; Bailey & Piliouras [2019]. This is a critical failure of descent methods, but one which Daskalakis et al. [2018] showed can be overcome through "*optimism*", interpreted in this context as an "extra-gradient" step that pushes the training process further along the incumbent gradient – as a result, optimistic gradient descent (OGD) succeeds in cases where vanilla gradient descent (GD) fails (specifically, unconstrained bilinear saddle-point problems).

A common theme in the above is that, to obtain a principled methodology for training GANs, it is beneficial to first establish improvements in a more restricted setting, and then test whether these gains carry over to more demanding learning environments. Following these theoretical breadcrumbs, we focus on a class of non-monotone problems whose solutions are related to those of a naturally associated variational inequality, a property which we call *coherence*. Then, hoping to overcome the shortcomings of ordinary descent methods by exploiting the problem's geometry, we examine the convergence of MD in coherent problems. On the positive side, we show that if a problem is *strictly* coherent (a condition satisfied by all strictly convex-concave problems), MD converges almost surely, even in stochastic problems (Theorem 3.1). However, under *null coherence* (the "saturated" opposite to strict coherence), MD spirals outwards from the problem's solutions and may cycle in perpetuity. The null coherence property covers all bilinear models, so this result encompasses fully the analysis of Bailey & Piliouras [2018] for GD and follow-the-regularized-leader (FTRL) in general bilinear zero-sum games within our coherence framework. Thus, in and by themselves, gradient/mirror descent methods *do not* suffice for training convoluted, adversarial deep learning models.

To mitigate this deficiency, we consider the addition of an extra-gradient step which looks ahead and takes an additional step along a "future" gradient. This technique was first introduced by Korpelevich [1976] and subsequently gained great popularity as the basis of the mirror-prox algorithm

of Nemirovski [2004] which achieves an optimal $\mathcal{O}(1/n)$ convergence rate in Lipschitz monotone variational inequalities (see also Nesterov, 2007, for a primal-dual variant of the method and Juditsky et al., 2011, for an extension to stochastic variational inequalities and saddle-point problems).

In the learning literature, the extra-gradient technique (or, sometimes, a variant thereof) is often referred to as *optimistic mirror descent* (OMD) [Rakhlin & Sridharan, 2013] and its effectiveness in GAN training was recently examined by Daskalakis et al. [2018] and Yadav et al. [2018] (the latter involving a damping mechanism for only one of the players). More recently, Gidel et al. [2018] considered a variant method which incorporates a mechanism that "extrapolates from the past" in order to circumvent the need for a second oracle call in the extra-gradient step. Specifically, Gidel et al. [2018] showed that the extra-gradient algorithm with gradient reuse converges *a)* geometrically in strongly monotone, deterministic variational inequalities; and *b)* ergodically in general stochastic variational inequalities, achieving in that case an oracle complexity bound that is $\sqrt{13/7}/2 \approx 68\%$ of a bound previously established by Juditsky et al. [2011] for the mirror-prox algorithm.

However, beyond convex-concave problems, averaging offers no tangible benefits because there is no way to relate the value of the ergodic average to the value of the iterates. As a result, moving closer to GAN training requires changing both the algorithm's output as well as the accompanying analysis. With this as our guiding principle, we first show that the *last iterate* of OMD converges in all coherent problems, *including null-coherent ones*. As a special case, this generalizes and extends the results of Noor et al. [2011] for OGD in pseudo-monotone problems, and also settles in the affirmative an open question of Daskalakis et al. [2018] concerning the convergence of the last iterate of OGD in nonlinear problems. Going beyond deterministic problems, we also show that OMD converges with probability 1 even in *stochastic* saddle-point problems that are strictly coherent. These results complement the existing literature on the topic by showing that a cheap extra-gradient add-on can lead to significant performance gains when applied to state-of-the-art methods (such as Adam). We validate this prediction for a wide array of standard GAN models in Section 5.

## 2 PROBLEM SETUP AND PRELIMINARIES

**Saddle-point problems.** Consider a saddle-point problem of the general form

$$\min_{x_1 \in \mathcal{X}_1} \max_{x_2 \in \mathcal{X}_2} f(x_1, x_2), \tag{SP}$$

where each feasible region $\mathcal{X}_i$, $i = 1, 2$, is a compact convex subset of a finite-dimensional normed space $\mathcal{V}_i \equiv \mathbb{R}^{d_i}$, and $f: \mathcal{X} \equiv \mathcal{X}_1 \times \mathcal{X}_2 \to \mathbb{R}$ denotes the problem's value function. From a game-theoretic standpoint, (SP) can be seen as a *zero-sum game* between two optimizing agents (or *players*): Player 1 (the *minimizer*) seeks to incur the least possible loss, while Player 2 (the *maximizer*) seeks to obtain the highest possible reward – both determined by $f(x_1, x_2)$.

To solve (SP), we will focus on incremental processes that exploit the individual loss/reward gradients of $f$ (assumed throughout to be at least $C^1$-smooth). Since the individual gradients of $f$ will play a key role in our analysis, we will encode them in a single vector as

$$g(x) = (g_1(x), g_2(x)) = (\nabla_{x_1} f(x_1, x_2), -\nabla_{x_2} f(x_1, x_2)) \tag{2.1}$$

and, following standard conventions, we will treat $g(x)$ as an element of $\mathcal{Y} \equiv \mathcal{V}^*$, the dual of the ambient space $\mathcal{V} \equiv \mathcal{V}_1 \times \mathcal{V}_2$, assumed to be endowed with the product norm $\|x\|^2 = \|x_1\|^2 + \|x_2\|^2$.

**Variational inequalities and coherence.** Most of the literature on saddle-point problems has focused on the *monotone* case, i.e., when $f$ is *convex-concave*. In such problems, it is well known that solutions of (SP) can be characterized equivalently as solutions of the Stampacchia variational inequality

$$\langle g(x^*), x - x^* \rangle \geq 0 \quad \text{for all } x \in \mathcal{X} \tag{SVI}$$

or, in Minty form:

$$\langle g(x), x - x^* \rangle \geq 0 \quad \text{for all } x \in \mathcal{X}. \tag{MVI}$$

The equivalence between solutions of (SP), (SVI) and (MVI) extends well beyond the realm of monotone problems: it trivially includes all bilinear problems ($f(x_1, x_2) = x_1^\top M x_2$), pseudo-monotone objectives as in Noor et al. [2011], etc. For a concrete example which is not even pseudo-monotone, consider the problem

$$\min_{x_1 \in [-1,1]} \max_{x_2 \in [-1,1]} (x_1^4 x_2^2 + x_1^2 + 1)(x_1^2 x_2^4 - x_2^2 + 1). \tag{2.2}$$

The only saddle-point of $f$ is $x^* = (0, 0)$: it is easy to check that $x^*$ is also the unique solution of the corresponding variational inequality (VI) problems, despite the fact that $f$ is not even pseudo-monotone.[1] This shows that this equivalence encompasses a wide range of phenomena that are innately incompatible with convexity/monotonicity, even in the lowest possible dimension; for an in-depth discussion we refer the reader to Facchinei & Pang [2003].

Motivated by all this, we introduce below the following notion of *coherence*:

**Definition 2.1.** We say that (SP) is *coherent* if:

1. Every solution of (SVI) also solves (SP).

2. There exists a solution $p$ of (SP) that satisfies (MVI).

3. Every solution $x^*$ of (SP) satisfies (MVI) locally, i.e., for all $x$ sufficiently close to $x^*$.

In the above, if (MVI) holds as a strict inequality whenever $x$ is not a solution thereof, (SP) will be called *strictly coherent*; by contrast, if (MVI) holds as an equality for all $x \in \mathcal{X}$, we will say that (SP) is *null-coherent*.

The notion of coherence will play a central part in our considerations, so a few remarks are in order. To the best of our knowledge, its first antecedent is a gradient condition examined by Bottou [1998] in the context of nonlinear programming; we borrow the term "coherence" from the more recent paper of Zhou et al. [2017b] who used the term "variational coherence" for a stronger variant of the above definition.

We should also note here that the set of solutions of a coherent problem does not need to be convex: for instance, if Player 1 controls $(x, y)$, and the objective function is $f(x, y) = x^2 y^2$ (i.e., Player 2 has no impact in the game), the set of solutions is the non-convex set $\mathcal{X}^* = \{(x, y) : x = 0 \text{ or } y = 0\}$. Moreover, regarding the distinction between coherence and *strict* coherence, we show in Appendix A that (SP) is strictly coherent when $f$ is strictly convex-concave. At the other end of the spectrum, typical examples of problems that are null-coherent are bilinear objectives with an interior solution: for instance, $f(x_1, x_2) = x_1 x_2$ with $x_1, x_2 \in [-1, 1]$ has $\langle g(x), x \rangle = x_1 x_2 - x_2 x_1 = 0$ for all $x_1, x_2 \in [-1, 1]$, so it is null-coherent. Finally, neither strict, nor null coherence imply a unique solution to (SP), a property which is particularly relevant for GANs (the first example above is strictly coherent, but does not admit a unique solution).

## 3 MIRROR DESCENT

**The method.** Motivated by its prolific success in convex programming, our starting point will be the well-known *mirror descent* (MD) method of Nemirovski & Yudin [1983], suitably adapted to our saddle-point context. Several variants of the method exist, ranging from dual averaging [Nesterov, 2009] to follow-the-regularized-leader; for a survey, we refer the reader to Bubeck [2015].

The basic idea of mirror descent is to generate a new state variable $x^+$ from some starting state $x$ by taking a "mirror step" along a gradient-like vector $y$. To do this, let $h : \mathcal{X} \to \mathbb{R}$ be a continuous and $K$-strongly convex *distance-generating function* (DGF) on $\mathcal{X}$, i.e.,

$$h(tx + (1 - t)x') \le th(x) + (1 - t)h(x') - \tfrac{1}{2}Kt(1 - t)\|x' - x\|^2, \qquad (3.1)$$

for all $x, x' \in \mathcal{X}$ and all $t \in [0, 1]$. In terms of smoothness (and in a slight abuse of notation), we also assume that the subdifferential of $h$ admits a *continuous selection*, i.e., a continuous function $\nabla h : \operatorname{dom} \partial h \to \mathcal{Y}$ such that $\nabla h(x) \in \partial h(x)$ for all $x \in \operatorname{dom} \partial h$.[2] Then, following Bregman [1967], $h$ generates a pseudo-distance on $\mathcal{X}$ via the relation

$$D(p, x) = h(p) - h(x) - \langle \nabla h(x), p - x \rangle \quad \text{for all } p \in \mathcal{X}, x \in \operatorname{dom} \partial h. \qquad (3.2)$$

This pseudo-distance is known as the *Bregman divergence*. As we show in Appendix B, we have $D(p, x) \ge \tfrac{1}{2}K\|x - p\|^2$, so the convergence of a sequence $X_n$ to some target point $p$ can be verified by showing that $D(p, X_n) \to 0$. On the other hand, $D(p, x)$ typically fais to be symmetric and/or satisfy

---

[1]To see this, simply note that $f(x_1, x_2)$ is *multi-modal* in $x_2$ for certain values of $x_1$ (e.g., when $x_1 = 1/2$).

[2]Recall here that the *subdifferential* of $h$ at $x \in \mathcal{X}$ is defined as $\partial h(x) \equiv \{y \in \mathcal{Y} : h(x') \ge h(x) + \langle y, x' - x \rangle$ for all $x \in \mathcal{V}\}$, with the standard convention $h(x) = \infty$ for all $x \in \mathcal{V} \setminus \mathcal{X}$.

---

**Algorithm 1:** mirror descent (MD) for saddle-point problems

```
Require: K-strongly convex regularizer h: X → ℝ, step-size sequence γₙ > 0
1: choose X ∈ dom ∂h                                    #initialization
2: for n = 1, 2, ... do
3:    oracle query at X returns g                       #gradient feedback
4:    set X ← Pₓ(−γₙg)                                  #new state
5: end for
6: return X
```

---

the triangle inequality, so it is not a true distance function per se. Moreover, the level sets of $D(p, x)$ may fail to form a neighborhood basis of $p$, so the convergence of $X_n$ to $p$ does not necessarily imply that $D(p, X_n) \to 0$; we provide an example of this behavior in Appendix B. For technical reasons, it will be convenient to assume that such phenomena do not occur, i.e., that $D(p, X_n) \to 0$ whenever $X_n \to p$. This mild regularity condition is known in the literature as "Bregman reciprocity" [Chen & Teboulle, 1993; Alvarez et al., 2004; Mertikopoulos & Staudigl, 2018; Bravo et al., 2018], and it will be our standing assumption in what follows (note also that it holds trivially for both Examples 3.1 and 3.2 below).

Now, as with standard Euclidean distances, the Bregman divergence generates an associated *prox-mapping* defined as

$$P_x(y) = \arg\min_{x' \in \mathcal{X}} \{\langle y, x - x' \rangle + D(x', x)\} \quad \text{for all } x \in \text{dom } \partial h, y \in \mathcal{Y}. \tag{3.3}$$

In analogy with the Euclidean case (discussed below), the prox-mapping (3.3) produces a feasible point $x^+ = P_x(y)$ by starting from $x \in \text{dom } \partial h$ and taking a step along a dual (gradient-like) vector $y \in \mathcal{Y}$. In this way, we obtain the *mirror descent* (MD) algorithm

$$X_{n+1} = P_{X_n}(-\gamma_n \hat{g}_n), \tag{MD}$$

where $\gamma_n$ is a variable step-size sequence and $\hat{g}_n$ is the calculated value of the gradient vector $g(X_n)$ at the $n$-th stage of the algorithm (for a pseudocode implementation, see Section 3).

For concreteness, two widely used examples of prox-mappings are as follows:

*Example* 3.1 (Euclidean projections). When $\mathcal{X}$ is endowed with the $L^2$ norm $\|\cdot\|_2$, the archetypal prox-function is the (square of the) norm itself, i.e., $h(x) = \frac{1}{2}\|x\|_2^2$. In that case, $D(p, x) = \frac{1}{2}\|x - p\|^2$ and the induced prox-mapping is

$$P_x(y) = \Pi(x + y), \tag{3.4}$$

with $\Pi(x) = \arg\min_{x' \in \mathcal{X}} \|x' - x\|^2$ denoting the ordinary Euclidean projection onto $\mathcal{X}$.

*Example* 3.2 (Entropic regularization). When $\mathcal{X}$ is a $d$-dimensional simplex, a widely used DGF is the (negative) Gibbs–Shannon entropy $h(x) = \sum_{j=1}^{d} x_j \log x_j$. This function is 1-strongly convex with respect to the $L^1$ norm [Shalev-Shwartz, 2011] and the associated pseudo-distance is the Kullback–Leibler divergence $D_{\text{KL}}(p, x) = \sum_{j=1}^{d} p_j \log(p_j/x_j)$; in turn, this yields the prox-mapping

$$P_x(y) = \frac{(x_j \exp(y_j))_{j=1}^{d}}{\sum_{j=1}^{d} x_j \exp(y_j)} \quad \text{for all } x \in \mathcal{X}^\circ, y \in \mathcal{Y}. \tag{3.5}$$

The update rule $x \leftarrow P_x(y)$ is known in the literature as the *multiplicative weights* (MW) algorithm [Arora et al., 2012], and is one of the centerpieces for learning in games [Fudenberg & Levine, 1998; Freund & Schapire, 1999; Cohen et al., 2017], adversarial bandits [Auer et al., 1995], etc.

Regarding the gradient input sequence $\hat{g}_n$ of (MD), we assume that it is obtained by querying a *first-order oracle* which outputs an estimate of $g(X_n)$ when called at $X_n$. This oracle could be either *perfect*, returning $\hat{g}_n = g(X_n)$ for all $n$, or *imperfect*, providing noisy gradient estimations.[3] By that token, we will make the following standard assumptions for the gradient feedback sequence $\hat{g}_n$ [Nesterov, 2007; Nemirovski et al., 2009; Juditsky et al., 2011]:

    a) *Unbiasedness:*               $\mathbb{E}[\hat{g}_n \mid \mathcal{F}_n] = g(X_n)$.

    b) *Finite mean square error:*    $\mathbb{E}[\|\hat{g}_n\|_*^2 \mid \mathcal{F}_n] \leq G^2$   for some finite $G \geq 0$.          (3.6)

---

[3]The reason for this is that, depending on the application at hand, gradients might be difficult to compute directly e.g., because they require huge amounts of data, the calculation of an unknown expectation, etc.

In the above, $\|y\|_* \equiv \sup\{\langle y, x \rangle : x \in \mathcal{V}, \|x\| \leq 1\}$ denotes the dual norm on $\mathcal{Y}$ while $\mathcal{F}_n$ represents the history (natural filtration) of the generating sequence $X_n$ up to stage $n$ (inclusive). Since $\hat{g}_n$ is generated randomly from $X_n$ at stage $n$, it is obviously not $\mathcal{F}_n$-measurable, i.e., $\hat{g}_n = g(X_n) + U_{n+1}$, where $U_n$ is an adapted martingale difference sequence with $\mathbb{E}[\|U_{n+1}\|_*^2 \mid \mathcal{F}_n] \leq \sigma^2$ for some finite $\sigma \geq 0$. Clearly, when $\sigma = 0$, we recover the exact gradient feedback framework $\hat{g}_n = g(X_n)$.

**Convergence analysis.** When (SP) is convex-concave, it is customary to take as the output of (MD) the so-called *ergodic average*

$$\bar{X}_n = \frac{\sum_{k=1}^n \gamma_k X_k}{\sum_{k=1}^n \gamma_k}, \tag{3.7}$$

or some other average of the sequence $X_n$ where the objective is sampled. The reason for this is that convexity guarantees – via Jensen's inequality and gradient monotonicity – that a regret-based analysis of (MD) can lead to explicit rates for the convergence of $\bar{X}_n$ to the solution set of (SP) [Nemirovski, 2004; Nesterov, 2007]. However, when the problem is not convex-concave, the standard proof techniques for establishing convergence of the method's ergodic average no longer apply; instead, we need to examine the convergence properties of the generating sequence $X_n$ of (MD) directly. With all this in mind, our main result for (MD) may be stated is as follows:

**Theorem 3.1.** *Suppose that* (MD) *is run with a gradient oracle satisfying* (3.6) *and a variable step-size sequence* $\gamma_n$ *such that* $\sum_{n=1}^{\infty} \gamma_n = \infty$. *Then:*

a) *If $f$ is strictly coherent and* $\sum_{n=1}^{\infty} \gamma_n^2 < \infty$, $X_n$ *converges (a.s.) to a solution of* (SP).

b) *If $f$ is null-coherent, the sequence* $\mathbb{E}[D(x^*, X_n)]$ *is non-decreasing for every solution $x^*$ of* (SP).

This result establishes an important dichotomy between strict and null coherence: *in strictly coherent problems, $X_n$ is attracted to the solution set of* (SP)*; in null-coherent problems, $X_n$ drifts away and cycles without converging.* In particular, this dichotomy leads to the following immediate corollaries:

**Corollary 3.2.** *Suppose that $f$ is strictly convex-concave. Then, with assumptions as above, $X_n$ converges (a.s.) to the (necessarily unique) solution of* (SP).

**Corollary 3.3.** *Suppose that $f$ is bilinear and admits an interior saddle-point $x^* \in \mathcal{X}^\circ$. If $X_1 \neq x^*$ and* (MD) *is run with exact gradient input* ($\sigma = 0$)*, we have* $\lim_{n \to \infty} D(x^*, X_n) > 0$.

Since bilinear models include all finite two-player, zero-sum games, Corollary 3.3 also encapsulates the non-convergence results of Daskalakis et al. [2018] and Bailey & Piliouras [2018] for gradient descent and FTRL respectively (for a more comprehensive formulation, see Proposition C.3 in Appendix C). The failure of (MD) to converge in this case is due to the fact that, witout a mitigating mechanism in place, a "blind" first-order step could overshoot and spiral outwards, even with a *vanishing* step-size. This becomes even more pronounced in GANs where it can lead to mode collapse and/or cycles between different modes; the next two sections address precisely these issues.

## 4 EXTRA-GRADIENT ANALYSIS

**The method.** In convex-concave problems, taking an average of the algorithm's generated samples as in (3.7) may resolve cycling phenomena by inducing an auxiliary sequence that gravitates towards the "center of mass" of the driving sequence $X_n$ (which orbits interior solutions). However, this technique cannot be employed in problems that are not convex-concave because the structure of $f$ cannot be leveraged to establish convergence of the ergodic average of the process.

In view of this, we replace averaging with an optimistic "extra-gradient" step which uses the obtained information to amortize the next prox step (possibly by exiting the convex hull of generated states). The seed of this "extra-gradient" idea dates back to Korpelevich [1976] and Nemirovski [2004], and has since found wide applications in optimization theory and beyond – for a survey, see Bubeck [2015] and references therein.

In a nutshell, given a state $x$, the extra-gradient method first generates an intermediate, "waiting" state $\hat{x} = P_x(-\gamma g(x))$ by taking a prox step as usual. However, instead of continuing from $\hat{x}$, the method samples $g(\hat{x})$ and goes back to the *original* state $x$ in order to generate a new state $x^+ = P_x(-\gamma g(\hat{x}))$. Based on this heuristic, we obtain the *optimistic mirror descent* (OMD) algorithm

$$\begin{aligned} X_{n+1/2} &= P_{X_n}(-\gamma_n \hat{g}_n) \\ X_{n+1} &= P_{X_n}(-\gamma_n \hat{g}_{n+1/2}) \end{aligned} \tag{OMD}$$

---

**Algorithm 2:** optimistic mirror descent (OMD) for saddle-point problems

---

```
Require: K-strongly convex regularizer h: X → R, step-size sequence γₙ > 0
1: choose X ∈ dom ∂h                                          #initialization
2: for n = 1, 2, ... do
3:     oracle query at X returns g                            #gradient feedback
4:     set X⁺ ← P_X(-γₙg)                                     #waiting state
5:     oracle query at X⁺ returns g⁺                          #gradient feedback
6:     set X ← P_X(-γₙg⁺)                                     #new state
7: end for
8: return X
```

---

where, in obvious notation, $\hat{g}_n$ and $\hat{g}_{n+1/2}$ represent gradient oracle queries at the incumbent and intermediate states $X_n$ and $X_{n+1/2}$ respectively. For a pseudocode implementation, see Algorithm 2; see also Rakhlin & Sridharan [2013] and Daskalakis et al. [2018] for a variant of the method with a "momentum" step, and Gidel et al. [2018] for a gradient reuse mechanism that replaces a second oracle call with a past gradient.

**Convergence analysis.**    In his original analysis, Nemirovski [2004] considered the ergodic average (3.7) of the algorithm's iterates and established an $\mathcal{O}(1/n)$ convergence rate in monotone problems. However, as we explained above, even though this kind of averaging is helpful in convex-concave problems, it does not provide any tangible benefits beyond this class: in more general problems, $X_n$ appears to be the most natural solution candidate.

Our first result below justifies this choice in the class of coherent problems:

**Theorem 4.1.** *Suppose that* (SP) *is coherent and g is L-Lipschitz continuous. If* (OMD) *is run with exact gradient input* ($\sigma = 0$) *and* $\gamma_n$ *such that* $0 < \inf_n \gamma_n \leq \sup_n \gamma_n < K/L$, *the sequence* $X_n$ *converges monotonically to a solution* $x^*$ *of* (SP), *i.e.,* $D(x^*, X_n)$ *decreases monotonically to 0.*

**Corollary 4.2.** *Suppose that f is bilinear. If* (OMD) *is run with assumptions as above, the sequence* $X_n$ *converges monotonically to a solution of* (SP).

Theorem 4.1 includes as a special case the analysis of Facchinei & Pang [2003, Theorem 12.1.11] for optimistic gradient descent and the more recent results of Noor et al. [2011] for pseudo-monotone problems. Importantly, Theorem 4.1 shows that the extra-gradient step plays a crucial role in stabilizing (MD): not only does (OMD) converge in problems where (MD) provably fails, but this convergence is, in fact, monotonic. In other words, at each iteration, (OMD) comes closer to a solution of (SP), whereas (MD) may spiral outwards, ultimately converging to a limit cycle. This phenomenon is seen clearly in Fig. 1, and also in the detailed analysis of Appendix C.

Of course, except for very special cases, the monotonic convergence of $X_n$ cannot hold when the gradient input to (OMD) is imperfect: a single "bad" sample of $\hat{g}_n$ would suffice to throw $X_n$ off-track. In this case, we have:

**Theorem 4.3.** *Suppose that* (SP) *is strictly coherent and* (OMD) *is run with a gradient oracle satisfying* (3.6) *and a variable step-size sequence* $\gamma_n$ *such that* $\sum_{n=1}^{\infty} \gamma_n = \infty$ *and* $\sum_{n=1}^{\infty} \gamma_n^2 < \infty$. *Then, with probability 1,* $X_n$ *converges to a solution of* (SP).

It is worth noting here that the step-size policy in Theorem 4.3 is different than that of Theorem 4.1. This is due to *a)* the lack of randomness (which obviates the summability requirement $\sum_{n=1}^{\infty} \gamma_n^2 < \infty$ in Theorem 4.1); and *b)* the lack of Lipschitz continuity assumption (which, in the case of Theorem 4.1 guarantees monotonic decrease at each step, provided the step-size is not too big). Importantly, the maximum allowable step-size is also controlled by the strong convexity modulus of $h$, suggesting that the choice of distance-generating function can be fine-tuned further to allow for more aggressive step-size policies – a key benefit of mirror descent methods.

## 5    EXPERIMENTAL RESULTS

**Gaussian mixture models.**    For the experimental validation of our theoretical results, we began by evaluating the extra-gradient add-on in a highly multi-modal mixture of 16 Gaussians arranged in a $4 \times 4$ grid as in Metz et al. [2017]. The generator and discriminator have 6 fully connected layers with 384 neurons and Relu activations (plus an additional layer for data space projection), and the

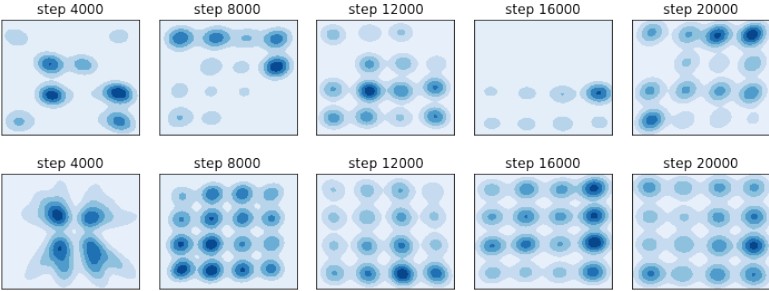

**(a)** Vanilla versus optimistic RMS (top and bottom respectively; $\gamma = 3 \times 10^{-4}$ in both cases).

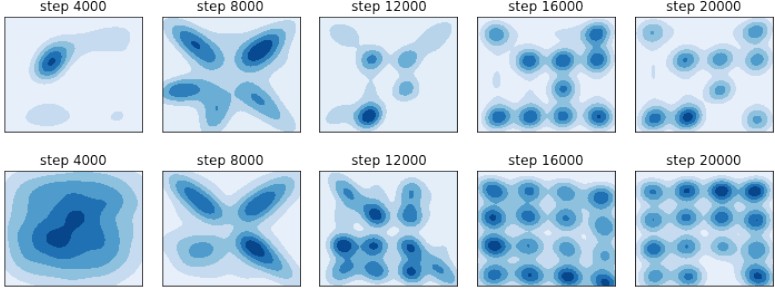

**(b)** Vanilla versus optimistic Adam (top and bottom respectively; $\gamma = 4 \times 10^{-5}$ in both cases).

**Figure 2:** Different algorithmic benchmarks (RMSprop and Adam): adding an extra-gradient step allows the training method to accurately learn the target data distribution and eliminates cycling and oscillatory instabilities.

generator generates 2-dimensional vectors. The output after {4000, 8000, 12000, 16000, 20000} iterations is shown in Fig. 2. The networks were trained with RMSprop [Tieleman & Hinton, 2012] and Adam [Kingma & Ba, 2014], and the results are compared to the corresponding extra-gradient variant (for an explicit pseudocode representation in the case of Adam, see Daskalakis et al. [2018] and Appendix E). Learning rates and hyperparameters were chosen by an inspection of grid search results so as to enable a fair comparison between each method and its look-ahead version. Overall, the different optimization strategies without look-ahead exhibit mode collapse or oscillations throughout the training period (we ran all models for at least 20000 iterations in order to evaluate the hopping behavior of the generator). In all cases, the extra-gradient add-on performs consistently better in learning the multi-modal distribution and greatly reduces occurrences of oscillatory behavior.

**Experiments with standard datasets.** In our experiments with Gaussian mixture models (GMMs), the most promising training method was Adam with an extra-gradient step (a concrete pseudocode implementation is provided in Appendix E). Motivated by this, we trained a Wasserstein-GAN on the CelebA and CIFAR-10 datasets using Adam, both with and without an extra-gradient step. The architecture employed was a standard DCGAN; hyperparameters and network architecture details may be found in Appendix E. Subsequently, to quantify the gains of the extra-gradient step, we employed the widely used inception score and Fréchet distance metrics, for which we report the results in Fig. 3. Under both metrics, the extra-gradient add-on provides consistently higher scores after an initial warm-up period (and is considerably more stable). For visualization purposes, we also present in Fig. 4 an ensemble of samples generated at the end of the training period. Overall, the generated samples provide accurate feature representation and low distortion (especially in CelebA).

## 6 CONCLUSIONS

Our results suggest that the implementation of an optimistic, extra-gradient step is a flexible add-on that can be easily attached to a wide variety of GAN training methods (RMSProp, Adam, SGA, etc.), and provides noticeable gains in performance and stability. From a theoretical standpoint, the dichotomy between strict and null coherence provides a justification of why this is so: optimism eliminates cycles and, in so doing, stabilizes the method. We find this property particularly appealing

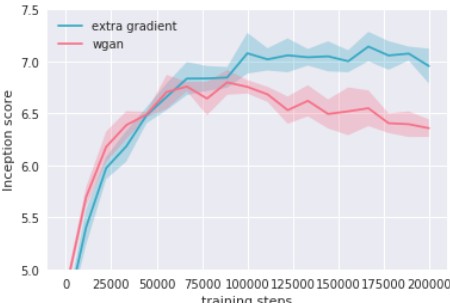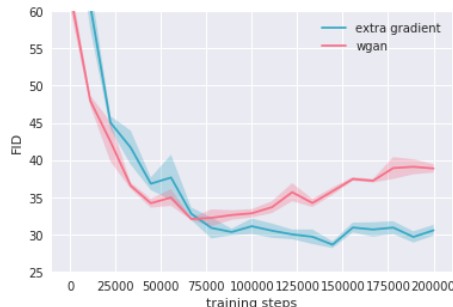

**Figure 3:** Left: Inception score (left) and Fréchet distance (right) on CIFAR-10 when training with Adam (with and without an extra-gradient step). Results are averaged over 8 sample runs with different random seeds.

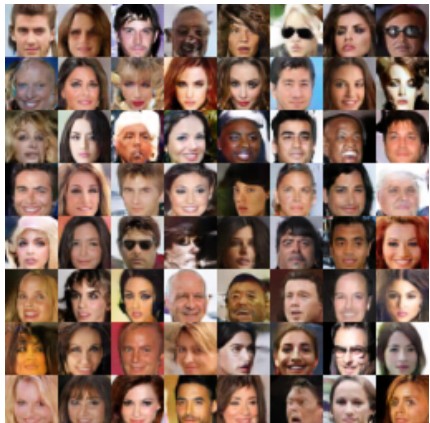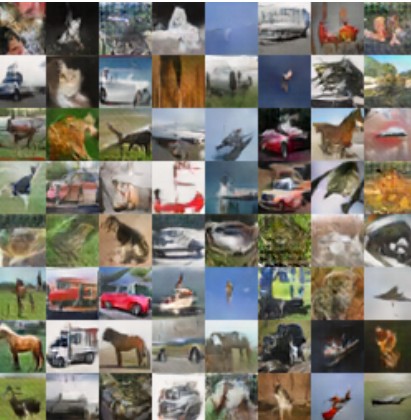

**Figure 4:** Samples generated by Adam with an extra-gradient step on CelebA (left) and CIFAR-10 (right).

because it paves the way to a local analysis with provable convergence guarantees in multi-modal settings, and beyond zero-sum games; we intend to examine this question in future work.

ACKNOWLEDGMENTS

P. Mertikopoulos was partially supported by the French National Research Agency (ANR) grant ORACLESS (ANR–16–CE33–0004–01), the CNRS PEPS program under grant MixedGAN, and the FMJH PGMO program under grant HEAVY.NET. G. Piliouras was partially supported by SUTD grant SRG ESD 2015 097, MOE AcRF Tier 2 grant 2016-T2-1-170, grant PIE-SGP-AI-2018-01 and NRF fellowship NRF-NRFF2018-07. This work was partly funded by the deep learning 2.0 program at A*STAR.

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

## A    COHERENT SADDLE-POINT PROBLEMS

We begin our discussion with some basic results on coherence:

**Proposition A.1.** *If $f$ is convex-concave, (SP) is coherent. In addition, if $f$ is strictly convex-concave, (SP) is strictly coherent.*

*Proof.* Let $x^*$ be a solution point of (SP). Since $f$ is convex-concave, first-order optimality gives

$$\langle g_1(x_1^*, x_2^*), x_1 - x_1^* \rangle = \langle \nabla_{x_1} f(x_1^*, x_2^*), x_1 - x_1^* \rangle \geq 0, \tag{A.1a}$$

and

$$\langle g_2(x_1^*, x_2^*), x_2 - x_2^* \rangle = \langle -\nabla_{x_2} f(x_1^*, x_2^*), x_2 - x_2^* \rangle \geq 0. \tag{A.1b}$$

Combining the two, we readily obtain the (Stampacchia) variational inequality

$$\langle g(x^*), x - x^* \rangle \geq 0 \quad \text{for all } x \in \mathcal{X}. \tag{A.2}$$

In addition to the above, the fact that $f$ is convex-concave also implies that $g(x)$ is *monotone* in the sense that

$$\langle g(x') - g(x), x' - x \rangle \geq 0 \tag{A.3}$$

for all $x, x' \in \mathcal{X}$ [Bauschke & Combettes, 2017]. Thus, setting $x' \leftarrow x^*$ in (A.3) and invoking (SVI), we get

$$\langle g(x), x - x^* \rangle \geq \langle g(x^*), x - x^* \rangle \geq 0, \tag{A.4}$$

i.e., (MVI) is satisfied.

To establish the converse implication, focus for concreteness on the minimizer, and note that (MVI) implies that

$$\langle g_1(x), x_1 - x_1^* \rangle \geq 0 \quad \text{for all } x_1 \in \mathcal{X}_1. \tag{A.5}$$

Now, if we fix some $x_1 \in \mathcal{X}_1$ and consider the function $\phi(t) = f(x_1^* + t(x_1 - x_1^*), x_2^*)$, the inequality (A.5) yields

$$\phi'(t) = \langle g(x_1^* + t(x_1 - x_1^*), x_2^*), x_1 - x_1^* \rangle$$
$$= \frac{1}{t} \langle g(x_1^* + t(x_1 - x_1^*), x_2^*), x_1^* + t(x_1 - x_1^* - x_1^*) \rangle \geq 0, \tag{A.6}$$

for all $t \in [0, 1]$. This implies that $\phi$ is nondecreasing, so $f(x_1, x_2^*) = \phi(1) \geq \phi(0) = f(x_1^*, x_2^*)$. The maximizing component follows similarly, showing that $x^*$ is a solution of (SP) and, in turn, establishing that (SP) is coherent.

For the strict part of the claim, the same line of reasoning shows that if $\langle g(x), x - x^* \rangle = 0$ for some $x$ that is not a saddle-point of $f$, the function $\phi(t)$ defined above must be constant on $[0, 1]$, indicating in turn that $f$ cannot be strictly convex-concave, a contradiction. ∎

We proceed to show that the solution set of a coherent saddle-point problem is closed (we will need this regularity result in the convergence analysis of Appendix C):

**Lemma A.2.** *Let $\mathcal{X}^*$ denote the solution set of (SP). If (SP) is coherent, $\mathcal{X}^*$ is closed.*

*Proof.* Let $x_n^*$, $n = 1, 2, \ldots$, be a sequence of solutions of (SP) converging to some limit point $x^* \in \mathcal{X}$. To show that $\mathcal{X}^*$ is closed, it suffices to show that $x^* \in \mathcal{X}$.

Indeed, given that (SP) is coherent, every solution thereof satisfies (MVI), so we have $\langle g(x), x - x_n^* \rangle \geq 0$ for all $x \in \mathcal{X}$. With $x_n^* \to x^*$ as $n \to \infty$, it follows that

$$\langle g(x), x - x^* \rangle = \lim_{n \to \infty} \langle g(x), x - x_n^* \rangle \geq 0 \quad \text{for all } x \in \mathcal{X}, \tag{A.7}$$

i.e., $x^*$ satisfies (MVI). By coherence, this implies that $x^*$ is a solution of (SP), as claimed. ∎

## B    Properties of the Bregman divergence

In this appendix, we provide some auxiliary results and estimates that are used throughout the convergence analysis of Appendix C. Some of the results we present here (or close variants thereof) are not new [see e.g., Nemirovski et al., 2009; Juditsky et al., 2011]. However, the hypotheses used to obtain them vary wildly in the literature, so we provide all the necessary details for completeness.

To begin, recall that the Bregman divergence associated to a $K$-strongly convex distance-generating function $h\colon \mathcal{X} \to \mathbb{R}$ is defined as

$$D(p, x) = h(p) - h(x) - \langle \nabla h(x), p - x \rangle \tag{B.1}$$

with $\nabla h(x)$ denoting a continuous selection of $\partial h(x)$. The induced prox-mapping is then given by

$$P_x(y) = \arg\min_{x' \in \mathcal{X}}\{\langle y, x - x' \rangle + D(x', x)\}$$
$$= \arg\max_{x' \in \mathcal{X}}\{\langle y + \nabla h(x), x' \rangle - h(x')\} \tag{B.2}$$

and is defined for all $x \in \operatorname{dom} \partial h$, $y \in \mathcal{Y}$ (recall here that $\mathcal{Y} \equiv \mathcal{V}^*$ denotes the dual of the ambient vector space $\mathcal{V}$). In what follows, we will also make frequent use of the convex conjugate $h^*\colon \mathcal{Y} \to \mathbb{R}$ of $h$, defined as

$$h^*(y) = \max_{x \in \mathcal{X}}\{\langle y, x \rangle - h(x)\}. \tag{B.3}$$

By standard results in convex analysis [Rockafellar, 1970, Chap. 26], $h^*$ is differentiable on $\mathcal{Y}$ and its gradient satisfies the identity

$$\nabla h^*(y) = \arg\max_{x \in \mathcal{X}}\{\langle y, x \rangle - h(x)\}. \tag{B.4}$$

For notational convenience, we will also write

$$Q(y) = \nabla h^*(y) \tag{B.5}$$

and we will refer to $Q\colon \mathcal{Y} \to \mathcal{X}$ as the *mirror map* generated by $h$. All these notions are related as follows:

**Lemma B.1.** *Let $h$ be a distance-generating function on $\mathcal{X}$. Then, for all $x \in \operatorname{dom} \partial h$, $y \in \mathcal{Y}$, we have:*

*a)*  $x = Q(y) \iff y \in \partial h(x).$ \hfill (B.6a)

*b)*  $x^+ = P_x(y) \iff \nabla h(x) + y \in \partial h(x^+) \iff x^+ = Q(\nabla h(x) + y).$ \hfill (B.6b)

*Finally, if $x = Q(y)$ and $p \in \mathcal{X}$, we have*

$$\langle \nabla h(x), x - p \rangle \leq \langle y, x - p \rangle. \tag{B.7}$$

*Remark.* By (B.6b), we have $\partial h(x^+) \neq \varnothing$, i.e., $x^+ \in \operatorname{dom} \partial h$. As a result, the update rule $x \leftarrow P_x(y)$ is *well-posed*, i.e., it can be iterated in perpetuity.

*Proof of Lemma B.1.*  For (B.6a), note that $x$ solves (B.3) if and only if $y - \partial h(x) \ni 0$, i.e., if and only if $y \in \partial h(x)$. Similarly, comparing (B.2) with (B.3), it follows that $x^+$ solves (B.2) if and only if $\nabla h(x) + y \in \partial h(x^+)$, i.e., if and only if $x^+ = Q(\nabla h(x) + y)$.

For (B.7), by a simple continuity argument, it suffices to show that the inequality holds for interior $p \in \mathcal{X}^\circ$. To establish this, let

$$\phi(t) = h(x + t(p - x)) - [h(x) + \langle y, x + t(p - x) \rangle]. \tag{B.8}$$

Since $h$ is strongly convex and $y \in \partial h(x)$ by (B.6a), it follows that $\phi(t) \geq 0$ with equality if and only if $t = 0$. Since $\psi(t) = \langle \nabla h(x + t(p - x)) - y, p - x \rangle$ is a continuous selection of subgradients of $\phi$ and both $\phi$ and $\psi$ are continuous on $[0, 1]$, it follows that $\phi$ is continuously differentiable with $\phi' = \psi$ on $[0, 1]$. Hence, with $\phi$ convex and $\phi(t) \geq 0 = \phi(0)$ for all $t \in [0, 1]$, we conclude that $\phi'(0) = \langle \nabla h(x) - y, p - x \rangle \geq 0$, which proves our assertion. $\blacksquare$

We continue with some basic bounds on the Bregman divergence before and after a prox step. The basic ingredient for these bounds is a generalization of the (Euclidean) law of cosines which is known in the literature as the "three-point identity" [Chen & Teboulle, 1993]:

**Lemma B.2.** *Let h be a distance-generating function on $\mathcal{X}$. Then, for all $p \in \mathcal{X}$ and all $x, x' \in$ dom $\partial h$, we have*

$$D(p, x') = D(p, x) + D(x, x') + \langle \nabla h(x') - \nabla h(x), x - p \rangle. \quad (B.9)$$

*Proof.* By definition, we have:

$$\begin{aligned}
D(p, x') &= h(p) - h(x') - \langle \nabla h(x'), p - x' \rangle \\
D(p, x) &= h(p) - h(x) - \langle \nabla h(x), p - x \rangle \\
D(x, x') &= h(x) - h(x') - \langle \nabla h(x'), x - x' \rangle.
\end{aligned} \quad (B.10)$$

Our claim then follows by adding the last two lines and subtracting the first. ∎

With this identity at hand, we have the following series of upper and lower bounds:

**Proposition B.3.** *Let h be a K-strongly convex distance-generating function on $\mathcal{X}$, fix some $p \in \mathcal{X}$, and let $x^+ = P_x(y)$ for $x \in$ dom $\partial h$, $y \in \mathcal{Y}$. We then have:*

$$D(p, x) \geq \frac{K}{2} \|x - p\|^2. \quad (B.11a)$$

$$D(p, x^+) \leq D(p, x) - D(x^+, x) + \langle y, x^+ - p \rangle \quad (B.11b)$$

$$\leq D(p, x) + \langle y, x - p \rangle + \frac{1}{2K} \|y\|_*^2 \quad (B.11c)$$

*Proof of* (B.11a). By the strong convexity of $h$, we get

$$h(p) \geq h(x) + \langle \nabla h(x), p - x \rangle + \frac{K}{2} \|p - x\|^2 \quad (B.12)$$

so (B.11a) follows by gathering all terms involving $h$ and recalling the definition of $D(p, x)$. ∎

*Proof of* (B.11b) *and* (B.11c). By the three-point identity (B.9), we readily obtain

$$D(p, x) = D(p, x^+) + D(x^+, x) + \langle \nabla h(x) - \nabla h(x^+), x^+ - p \rangle. \quad (B.13)$$

In turn, this gives

$$\begin{aligned}
D(p, x^+) &= D(p, x) - D(x^+, x) + \langle \nabla h(x^+) - \nabla h(x), x^+ - p \rangle \\
&\leq D(p, x) - D(x^+, x) + \langle y, x^+ - p \rangle,
\end{aligned} \quad (B.14)$$

where, in the last step, we used (B.7) and the fact that $x^+ = P_x(y)$, so $\nabla h(x) + y \in \partial h(x^+)$. The above is just (B.11b), so the first part of our proof is complete.

For (B.11c), the bound (B.14) gives

$$D(p, x^+) \leq D(p, x) + \langle y, x - p \rangle + \langle y, x^+ - x \rangle - D(x^+, x). \quad (B.15)$$

Therefore, by Young's inequality [Rockafellar, 1970], we get

$$\langle y, x^+ - x \rangle \leq \frac{K}{2} \|x^+ - x\|^2 + \frac{1}{2K} \|y\|_*^2, \quad (B.16)$$

and hence

$$\begin{aligned}
D(p, x^+) &\leq D(p, x) + \langle y, x - p \rangle + \frac{1}{2K} \|y\|_*^2 + \frac{K}{2} \|x^+ - x\|^2 - D(x^+, x) \\
&\leq D(p, x) + \langle y, x - p \rangle + \frac{1}{2K} \|y\|_*^2,
\end{aligned} \quad (B.17)$$

with the last step following from Lemma B.1 applied to $x$ in place of $p$. ∎

The first part of Proposition B.3 shows that $X_n$ converges to $p$ if $D(p, X_n) \to 0$. However, as we mentioned in the main body of the paper, the converse may fail: in particular, we could have $\liminf_{n \to \infty} D(p, X_n) > 0$ even if $X_n \to p$. To see this, let $\mathcal{X}$ be the $L^2$ ball of $\mathbb{R}^d$ and take $h(x) = -\sqrt{1 - \|x\|_2^2}$. Then, a straightforward calculation gives

$$D(p, x) = \frac{1 - \langle p, x \rangle}{\sqrt{1 - \|x\|_2^2}} \quad (B.18)$$

whenever $\|p\|_2 = 1$. The corresponding level sets $L_c(p) = \{x \in \mathbb{R}^d : D(p, x) = c\}$ of $D(p, \cdot)$ are given by the equation

$$1 - \langle p, x \rangle = c \sqrt{1 - \|x\|_2^2}, \tag{B.19}$$

which admits $p$ as a solution for all $c \geq 0$ (so $p$ belongs to the closure of $L_c(p)$ even though $D(p, p) = 0$ by definition). As a result, under this distance-generating function, it is possible to have $X_n \to p$ even when $\liminf_{n\to\infty} D(p, X_n) > 0$ (simply take a sequence $X_n$ that converges to $p$ while remaining on the same level set of $D$). As we discussed in the main body of the paper, such pathologies are discarded by the Bregman reciprocity condition

$$D(p, X_n) \to 0 \quad \text{whenever} \quad X_n \to p. \tag{B.20}$$

This condition comes into play at the very last part of the proofs of Theorems 3.1 and 4.1; other than that, we will not need it in the rest of our analysis.

Finally, for the analysis of the OMD algorithm, we will need to relate prox steps taken along different directions:

**Proposition B.4.** *Let $h$ be a $K$-strongly convex distance-generating function on $\mathcal{X}$ and fix some $p \in \mathcal{X}$, $x \in \operatorname{dom} \partial h$. Then:*

*a) For all $y_1, y_2 \in \mathcal{Y}$, we have:*

$$\|P_x(y_2) - P_x(y_1)\| \leq \frac{1}{K}\|y_2 - y_1\|_*, \tag{B.21}$$

*i.e., $P_x$ is $(1/K)$-Lipschitz.*

*b) In addition, letting $x_1^+ = P_x(y_1)$ and $x_2^+ = P_x(y_2)$, we have:*

$$D(p, x_2^+) \leq D(p, x) + \langle y_2, x_1^+ - p \rangle + [\langle y_2, x_2^+ - x_1^+ \rangle - D(x_2^+, x)] \tag{B.22a}$$

$$\leq D(p, x) + \langle y_2, x_1^+ - p \rangle + \frac{1}{2K}\|y_2 - y_1\|_*^2 - \frac{K}{2}\|x_1^+ - x\|^2. \tag{B.22b}$$

*Proof.* We begin with the proof of the Lipschitz property of $P_x$. Indeed, for all $p \in \mathcal{X}$, (B.7) gives

$$\langle \nabla h(x_1^+) - \nabla h(x) - y_1, x_1^+ - p \rangle \leq 0, \tag{B.23a}$$

and

$$\langle \nabla h(x_2^+) - \nabla h(x) - y_2, x_2^+ - p \rangle \leq 0. \tag{B.23b}$$

Therefore, setting $p \leftarrow x_2^+$ in (B.23a), $p \leftarrow x_1^+$ in (B.23b) and rearranging, we obtain

$$\langle \nabla h(x_2^+) - \nabla h(x_1^+), x_2^+ - x_1^+ \rangle \leq \langle y_2 - y_1, x_2^+ - x_1^+ \rangle. \tag{B.24}$$

By the strong convexity of $h$, we also have

$$K\|x_2^+ - x_1^+\|^2 \leq \langle \nabla h(x_2^+) - \nabla h(x_1^+), x_2^+ - x_1^+ \rangle. \tag{B.25}$$

Hence, combining (B.24) and (B.25), we get

$$K\|x_2^+ - x_1^+\|^2 \leq \langle y_2 - y_1, x_2^+ - x_1^+ \rangle \leq \|y_2 - y_1\|_* \|x_2^+ - x_1^+\|, \tag{B.26}$$

and our assertion follows.

For the second part of our claim, the bound (B.11b) of Proposition B.3 applied to $x_2^+ = P_x(y_2)$ readily gives

$$D(p, x_2^+) \leq D(p, x) - D(x_2^+, x) + \langle y_2, x_2^+ - p \rangle$$
$$= D(p, x) + \langle y_2, x_1^+ - p \rangle + [\langle y_2, x_2^+ - x_1^+ \rangle - D(x_2^+, x)] \tag{B.27}$$

thus proving (B.22a). To complete our proof, note that (B.11b) with $p \leftarrow x_2^+$ gives

$$D(x_2^+, x_1^+) \leq D(x_2^+, x) + \langle y_1, x_1^+ - x_2^+ \rangle - D(x_1^+, x), \tag{B.28}$$

or, after rearranging,

$$D(x_2^+, x) \geq D(x_2^+, x_1^+) + D(x_1^+, x) + \langle y_1, x_2^+ - x_1^+ \rangle. \tag{B.29}$$

We thus obtain

$$\langle y_2, x_2^+ - x_1^+ \rangle - D(x_2^+, x) \leq \langle y_2 - y_1, x_2^+ - x_1^+ \rangle - D(x_2^+, x_1^+) - D(x_1^+, x)$$

$$\leq \frac{\|y_2 - y_1\|_*^2}{2K} + \frac{K}{2}\|x_2^+ - x_1^+\|^2 - \frac{K}{2}\|x_2^+ - x_1^+\|^2 - \frac{K}{2}\|x_1^+ - x\|^2$$

$$\leq \frac{1}{2K}\|y_2 - y_1\|_*^2 - \frac{K}{2}\|x_1^+ - x\|^2, \tag{B.30}$$

where we used Young's inequality and (B.11a) in the second inequality. The bound (B.22b) then follows by substituting (B.30) in (B.27). ∎

## C   Convergence analysis of mirror descent

We begin by recalling the definition of the mirror descent algorithm. With notation as in the previous section, the algorithm is defined via the recursive scheme

$$X_{n+1} = P_{X_n}(-\gamma_n \hat{g}_n), \tag{MD}$$

where $\gamma_n$ is a variable step-size sequence and $\hat{g}_n$ is the calculated value of the gradient vector $g(X_n)$ at the $n$-th stage of the algorithm. As we discussed in the main body of the paper, the gradient input sequence $\hat{g}_n$ of (MD) is assumed to satisfy the standard oracle assumptions

       *a) Unbiasedness:*       $\mathbb{E}[\hat{g}_n \,|\, \mathcal{F}_n] = g(X_n).$

       *b) Finite mean square:*    $\mathbb{E}[\|\hat{g}_n\|_*^2 \,|\, \mathcal{F}_n] \leq G^2$  for some finite $G \geq 0.$

where $\mathcal{F}_n$ represents the history (natural filtration) of the generating sequence $X_n$ up to stage $n$ (inclusive).

With this preliminaries at hand, our convergence proof for (MD) under strict coherence will hinge on the following results:

**Proposition C.1.** *Suppose that* (SP) *is coherent and* (MD) *is run with a gradient oracle satisfying* (3.6) *and a variable step-size* $\gamma_n$ *such that* $\sum_{n=1}^{\infty} \gamma_n^2 < \infty$. *If* $x^* \in \mathcal{X}$ *is a solution of* (SP), *the Bregman divergence* $D(x^*, X_n)$ *converges (a.s.) to a random variable* $D(x^*)$ *with* $\mathbb{E}[D(x^*)] < \infty$.

**Proposition C.2.** *Suppose that* (SP) *is strictly coherent and* (MD) *is run with a gradient oracle satisfying* (3.6) *and a step-size* $\gamma_n$ *such that* $\sum_{n=1}^{\infty} \gamma_n = \infty$ *and* $\sum_{n=1}^{\infty} \gamma_n^2 < \infty$. *Then, with probability 1, there exists a (possibly random) solution* $x^*$ *of* (SP) *such that* $\liminf_{n\to\infty} D(x^*, X_n) = 0$.

Proposition C.1 can be seen as a "dichotomy" result: it shows that the Bregman divergence is an asymptotic constant of motion, so (MD) either converges to a saddle-point $x^*$ (if $D(x^*) = 0$) or to some nonzero level set of the Bregman divergence (with respect to $x^*$). In this way, Proposition C.1 rules out more complicated chaotic or aperiodic behaviors that may arise in general – for instance, as in the analysis of Palaiopanos et al. [2017] for the long-run behavior of the multiplicative weights algorithm in two-player games. However, unless this limit value can be somehow predicted (or estimated) in advance, this result cannot be easily applied. This is the main role of Proposition C.2: it shows that (MD) admits a subsequence converging to a solution of (SP) so, by (B.20), the limit of $D(x^*, X_n)$ must be zero.

Our first step is to prove Proposition C.2. To do this, we first recall the following law of large numbers for $L^2$ martingales:

**Theorem** (Hall & Heyde, 1980, Theorem 2.18). *Let* $Y_n = \sum_{k=1}^{n} \zeta_k$ *be a martingale and* $\tau_n$ *a non-decreasing sequence such that* $\lim_{n\to\infty} \tau_n = \infty$. *Then,*

$$\lim_{n\to\infty} Y_n/\tau_n = 0 \quad (a.s.) \tag{C.1}$$

*on the set* $\sum_{n=1}^{\infty} \tau_n^{-2} \mathbb{E}[\zeta_n^2 \,|\, \mathcal{F}_{n-1}] < \infty$.

With this in place, we have:

*Proof of Proposition C.2.* We begin with the technical observation that the solution set $\mathcal{X}^*$ of (SP) is closed – and hence, compact (cf. Lemma A.2 in Appendix A). Clearly, if $\mathcal{X}^* = \mathcal{X}$, there is nothing to show; hence, without loss of generality, we may assume in what follows that $\mathcal{X}^* \neq \mathcal{X}$.

Assume now ad absurdum that, with positive probability, the sequence $X_n$ generated by (MD) admits no limit points in $\mathcal{X}^*$. Conditioning on this event, and given that $\mathcal{X}^*$ is compact, there exists a (nonempty) compact set $\mathcal{C} \subset \mathcal{X}$ such that $\mathcal{C} \cap \mathcal{X}^* = \emptyset$ and $X_n \in \mathcal{C}$ for all sufficiently large $n$. Moreover, letting $p$ be as in Definition 2.1, we have $\langle g(x), x - p \rangle > 0$ whenever $x \in \mathcal{C}$. Therefore, by the continuity of $g$ and the compactness of $\mathcal{X}^*$ and $\mathcal{C}$, there exists some $a > 0$ such that

$$\langle g(x), x - p \rangle \geq a \quad \text{for all } x \in \mathcal{C}. \tag{C.2}$$

To proceed, let $D_n = D(p, X_n)$. Then, by Proposition B.3, we have

$$D_{n+1} = D(p, P_{X_n}(-\gamma_n \hat{g}_n)) \leq D(p, X_n) - \gamma_n \langle \hat{g}_n, X_n - p \rangle + \frac{\gamma_n^2}{2K} \|\hat{g}_n\|^2$$

$$= D_n - \gamma_n \langle g(X_n), X_n - p \rangle - \gamma_n \langle U_{n+1}, X_n - p \rangle + \frac{\gamma_n^2}{2K} \|\hat{g}_n\|_*^2$$

$$\leq D_n + \gamma_n \xi_{n+1} + \frac{\gamma_n^2}{2K} \|\hat{g}_n\|_*^2, \tag{C.3}$$

where, in the last line, we set $U_{n+1} = \hat{g}_n - g(X_n)$, $\xi_{n+1} = -\langle U_{n+1}, X_n - p \rangle$, and we invoked the assumption that (SP) is coherent. Hence, telescoping (C.3) yields the estimate

$$D_{n+1} \leq D_1 - \sum_{k=1}^n \gamma_k \langle g(X_k), X_k - p \rangle + \sum_{k=1}^n \gamma_k \xi_{k+1} + \sum_{k=1}^n \frac{\gamma_k^2}{2K} \|\hat{g}_k\|_*^2. \tag{C.4}$$

Subsequently, letting $\tau_n = \sum_{k=1}^n \gamma_k$ and using (C.2), we obtain

$$D_{n+1} \leq D_1 - \tau_n \left[ a - \frac{\sum_{k=1}^n \gamma_k \xi_{k+1}}{\tau_n} - \frac{(2K)^{-1} \sum_{k=1}^n \gamma_k^2 \|\hat{g}_k\|_*^2}{\tau_n} \right]. \tag{C.5}$$

By the unbiasedness hypothesis of (3.6) for $U_n$, we have $\mathbb{E}[\xi_{n+1} \mid \mathcal{F}_n] = \langle \mathbb{E}[U_{n+1} \mid \mathcal{F}_n], X_n - p \rangle = 0$ (recall that $X_n$ is $\mathcal{F}_n$-measurable by construction). Moreover, since $U_n$ is bounded in $L^2$ and $\gamma_n$ is $\ell^2$ summable (by assumption), it follows that

$$\sum_{n=1}^\infty \gamma_n^2 \mathbb{E}[\xi_{n+1}^2 \mid \mathcal{F}_n] \leq \sum_{n=1}^\infty \gamma_n^2 \|X_n - p\|^2 \, \mathbb{E}[\|U_{n+1}\|_*^2 \mid \mathcal{F}_n]$$

$$\leq \operatorname{diam}(\mathcal{X})^2 \sigma^2 \sum_{n=1}^\infty \gamma_n^2 < \infty. \tag{C.6}$$

Therefore, by the law of large numbers for $L^2$ martingales stated above [Hall & Heyde, 1980, Theorem 2.18], we conclude that $\tau_n^{-1} \sum_{k=1}^n \gamma_k \xi_{k+1}$ converges to 0 with probability 1.

Finally, for the last term of (C.4), let $S_{n+1} = \sum_{k=1}^n \gamma_k^2 \|\hat{g}_k\|_*^2$. Since $\hat{g}_k$ is $\mathcal{F}_n$-measurable for all $k = 1, 2, \ldots, n-1$, we have

$$\mathbb{E}[S_{n+1} \mid \mathcal{F}_n] = \mathbb{E} \left[ \sum_{k=1}^{n-1} \gamma_k^2 \|\hat{g}_k\|_*^2 + \gamma_n^2 \|\hat{g}_n\|_*^2 \,\middle|\, \mathcal{F}_n \right] = S_n + \gamma_n^2 \mathbb{E}[\|\hat{g}_n\|_*^2 \mid \mathcal{F}_n] \geq S_n, \tag{C.7}$$

i.e., $S_n$ is a submartingale with respect to $\mathcal{F}_n$. Furthermore, by the law of total expectation, we also have

$$\mathbb{E}[S_{n+1}] = \mathbb{E}[\mathbb{E}[S_{n+1} \mid \mathcal{F}_n]] \leq G^2 \sum_{k=1}^n \gamma_n^2 \leq G^2 \sum_{k=1}^\infty \gamma_n^2 < \infty, \tag{C.8}$$

so $S_n$ is bounded in $L^1$. Hence, by Doob's submartingale convergence theorem [Hall & Heyde, 1980, Theorem 2.5], we conclude that $S_n$ converges to some (almost surely finite) random variable $S_\infty$ with $\mathbb{E}[S_\infty] < \infty$, implying in turn that $\lim_{n \to \infty} S_{n+1}/\tau_n = 0$ (a.s.).

Applying all of the above, the estimate (C.4) gives $D_{n+1} \leq D_1 - a\tau_n/2$ for sufficiently large $n$, so $D(p, X_n) \to -\infty$, a contradiction. Going back to our original assumption, this shows that, at least one of the limit points of $X_n$ must lie in $\mathcal{X}^*$ (a.s.), as claimed. ∎

We now turn to the proof of Proposition C.1:

*Proof of Proposition C.1.* Let $x^* \in \mathcal{X}^*$ be a limit point of $X_n$, as guaranteed by Proposition C.2, and let $D_n = D(x^*, X_n)$. Then, by Proposition B.3, we have

$$D_{n+1} = D(x^*, P_{X_n}(-\gamma_n \hat{g}_n)) \leq D(x^*, X_n) - \gamma_n \langle \hat{g}_n, X_n - x^* \rangle + \frac{\gamma_n^2}{2K} \|\hat{g}_n\|^2$$

$$= D_n - \gamma_n \langle g(X_n), X_n - x^* \rangle - \gamma_n \langle U_{n+1}, X_n - x^* \rangle + \frac{\gamma_n^2}{2K} \|\hat{g}_n\|_*^2 \tag{C.9}$$

and hence, for large enough $n$:

$$D_{n+1} \leq D_n + \gamma_n \xi_{n+1} + \frac{\gamma_n^2}{2K} \|\hat{g}_n\|_*^2, \tag{C.10}$$

where we used the ansatz that $\langle g(X_n), X_n - x^* \rangle \leq 0$ for sufficiently large $n$ (to be proved below), and, as in the proof of Proposition C.2, we set $U_{n+1} = \hat{g}_n - g(X_n)$, $\xi_{n+1} = -\langle U_{n+1}, X_n - x^* \rangle$. Thus, conditioning on $\mathcal{F}_n$ and taking expectations, we get

$$\mathbb{E}[D_{n+1} \,|\, \mathcal{F}_n] \leq D_n + \mathbb{E}[\xi_{n+1} \,|\, \mathcal{F}_n] + \frac{\gamma_n^2}{2K} \mathbb{E}[\|\hat{g}_n\|_*^2 \,|\, \mathcal{F}_n] \leq D_n + \frac{G^2}{2K}\gamma_n^2, \tag{C.11}$$

where we used the oracle assumptions (3.6) and the fact that $X_n$ is $\mathcal{F}_n$-measurable (by definition). Now, letting $R_n = D_n + (2K)^{-1}G^2 \sum_{k=n}^{\infty} \gamma_k^2$, the estimate (C.10) gives

$$\mathbb{E}[R_{n+1} \,|\, \mathcal{F}_n] = \mathbb{E}[D_{n+1} \,|\, \mathcal{F}_n] + \frac{G^2}{2K} \sum_{k=n+1}^{\infty} \gamma_k^2 \leq D_n + \frac{G^2}{2K} \sum_{k=n}^{\infty} \gamma_k^2 = R_n, \tag{C.12}$$

i.e., $R_n$ is an $\mathcal{F}_n$-adapted supermartingale. Since $\sum_{n=1}^{\infty} \gamma_n^2 < \infty$, it follows that

$$\mathbb{E}[R_n] = \mathbb{E}[\mathbb{E}[R_n \,|\, \mathcal{F}_{n-1}]] \leq \mathbb{E}[R_{n-1}] \leq \cdots \leq \mathbb{E}[R_1] \leq \mathbb{E}[D_1] + \frac{G^2}{2K} \sum_{n=1}^{\infty} \gamma_n^2 < \infty, \tag{C.13}$$

i.e., $R_n$ is uniformly bounded in $L^1$. Thus, by Doob's convergence theorem for supermartingales [Hall & Heyde, 1980, Theorem 2.5], it follows that $R_n$ converges (a.s.) to some finite random variable $R_\infty$ with $\mathbb{E}[R_\infty] < \infty$. In turn, by inverting the definition of $R_n$, this shows that $D_n$ converges (a.s.) to some random variable $D(x^*)$ with $\mathbb{E}[D(x^*)] < \infty$, as claimed.

It remains to be shown that $\langle g(X_n), X_n - x^* \rangle \geq 0$ for sufficiently large $n$. By Definition 2.1, this amounts to showing that, for all large enough $n$, $X_n$ lies in a neighborhood $U$ of $x^*$ such that (MVI) holds. Since $x^*$ has been chosen so that $\liminf D(x^*, X_n) = 0$, it follows that, for all $\varepsilon > 0$, there exists some $n_0$ such that $\sum_{n=n_0}^{\infty} \gamma_n^2 < \varepsilon$ and $X_{n_0} \in U$. Hence, arguing in the same way as in the proof of Theorem 5.2 of Zhou et al. [2017a], we conclude that $\mathbb{P}(X_n \in U$ for all $n \geq n_0) = 1$, implying in turn that $\langle g(X_n), X_n - x^* \rangle \geq 0$ for all $n \geq n_0$. This proves our last claim and concludes our proof. ∎

With all this at hand, we are finally in a position to prove our main result for (MD):

*Proof of Theorem 3.1(a).* Proposition C.2 shows that, with probability 1, there exists a (possibly random) solution $x^*$ of (SP) such that $\liminf_{n\to\infty}\|X_n - x^*\| = 0$ and, hence, $\liminf_{n\to\infty} D(x^*, X_n) = 0$ (by Bregman reciprocity). Since $\lim_{n\to\infty} D(x^*, X_n)$ exists with probability 1 (by Proposition C.1), it follows that $\lim_{n\to\infty} D(x^*, X_n) = \liminf_{n\to\infty} D(x^*, X_n) = 0$, i.e., $X_n$ converges to $x^*$. ∎

We proceed with the negative result hinted at in the main body of the paper, namely the failure of (MD) to converge under null coherence:

*Proof of Theorem 3.1(b).* The evolution of the Bregman divergence under (MD) satisfies the identity

$$\begin{aligned} D(x^*, X_{n+1}) &= D(x^*, X_n) + D(X_n, X_{n+1}) + \gamma_n \langle \hat{g}_n, X_n - x^* \rangle \\ &= D(x^*, X_n) + D(X_n, X_{n+1}) + \langle U_{n+1}, X_n - x^* \rangle \end{aligned} \tag{C.14}$$

where, in the last line, we used the null coherence assumption $\langle g(x), x - x^* \rangle = 0$ for all $x \in \mathcal{X}$. Since $D(X_n, X_{n+1}) \geq 0$, taking expecations above shows that $D(x^*, X_n)$ is nondecreasing, as claimed. ∎

With Theorem 3.1 at hand, the proof of Corollary 3.2 is an immediate consequence of the fact that strictly convex-concave problems satisfy strict coherence (Proposition A.1). As for Corollary 3.3, we provide below a more general result for two-player, zero-sum finite games.

To state it, let $\mathcal{A}_i = \{1, \ldots, A_i\}$, $i = 1, 2$, be two finite sets of *pure strategies*, and let $\mathcal{X}_i = \Delta(\mathcal{A}_i)$ denote the set of *mixed strategies* of player $i$. A *finite, two-player zero-sum game* is then defined by a matrix $M \in \mathbb{R}^{\mathcal{A}_1 \times \mathcal{A}_2}$ so that the loss of Player 1 and the reward of Player 2 in the mixed strategy profile $x = (x_1, x_2) \in \mathcal{X}$ are concurrently given by

$$f(x_1, x_2) = x_1^\top M x_2 \tag{C.15}$$

Then, writing $\Gamma \equiv \Gamma(\mathcal{A}_1, \mathcal{A}_2, M)$ for the resulting game, we have:

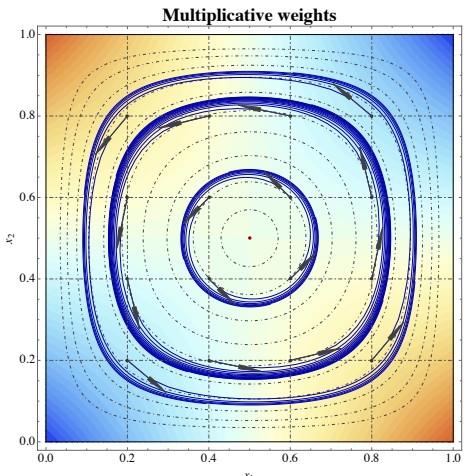
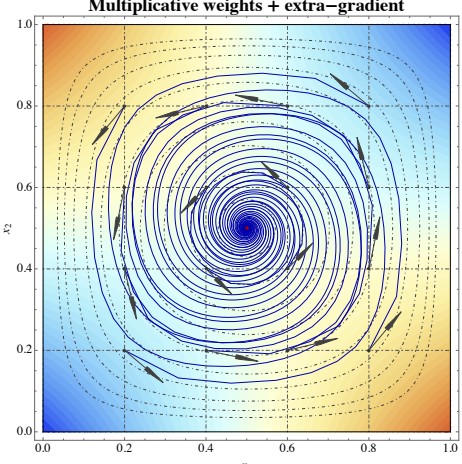

**Figure 5:** Trajectories of vanilla and optimistic mirror descent in a zero-sum game of Matching Pennies (left and right respectively). Colors represent the contours of the objective, $f(x_1, x_2) = (x_1 - 1/2)(x_2 - 1/2)$.

**Proposition C.3.** *Let $\Gamma$ be a two-player zero-sum game with an interior Nash equilibrium $x^*$. If $X_1 \neq x^*$ and (MD) is run with exact gradient input ($\sigma^2 = 0$), we have $\lim_{n \to \infty} D(x^*, X_n) > 0$. If, in addition, $\sum_{n=1}^{\infty} \gamma_n^2 < \infty$, $\lim_{n \to \infty} D(x^*, X_n)$ is finite.*

*Remark.* Note that non-convergence does not require any summability assumptions on $\gamma_n$.

In words, Proposition C.3 states that (MD) does not converge in finite zero-sum games with a unique interior equilibrium and exact gradient input: instead, $X_n$ cycles at positive Bregman distance from the game's Nash equilibrium. Heuristically, the reason for this behavior is that, for small $\gamma \to 0$, the incremental step $V_\gamma(x) = P_x(-\gamma g(x)) - x$ of (MD) is essentially tangent to the level set of $D(x^*, \cdot)$ that passes through $x$.[4] For finite $\gamma > 0$, things are even worse because $V_\gamma(x)$ points noticeably away from $x$, i.e., towards higher level sets of $D$. As a result, the "best-case scenario" for (MD) is to orbit $x^*$ (when $\gamma \to 0$); in practice, for finite $\gamma$, the algorithm takes small outward steps throughout its runtime, eventually converging to some limit cycle farther away from $x^*$.

We make this intuition precise below (for a schematic illustration, see also Fig. 1 above):

*Proof of Proposition C.3.* Write $v_1(x) = -Mx_2$ and $v_2(x) = x_1^\top M$ for the players' payoff vectors under the mixed strategy profile $x = (x_1, x_2)$. By construction, we have $g(x) = -(v_1(x), v_2(x))$. Furthermore, since $x^*$ is an interior equilibrium of $f$, elementary game-theoretic considerations show that $v_1(x^*)$ and $v_2(x^*)$ are both proportional to the constant vector of ones. We thus get

$$
\begin{aligned}
\langle g(x), x - x^* \rangle &= \langle v_1(x), x_1 - x_1^* \rangle + \langle v_2(x), x_2 - x_2^* \rangle \\
&= -x_1^\top M x_2 + (x_1^*)^\top M x_2 + x_1^\top M x_2 - x_1^\top M x_2^* \\
&= 0,
\end{aligned}
\tag{C.16}
$$

where, in the last line, we used the fact that $x^*$ is interior. This shows that $f$ satisfies null coherence, so our claim follows from Theorem 3.1(b).

For our second claim, arguing as above and using (B.11c), we get

$$
D(x^*, X_{n+1}) \leq D(x^*, X_n) + \gamma_n \langle g(X_n), X_n - x^* \rangle + \frac{\gamma_n^2}{2K} \|g(X_n)\|_*^2
$$

$$
\leq D(x^*, X_n) + \frac{\gamma_n^2 G^2}{2K}
\tag{C.17}
$$

with $G = \max_{x_1 \in \mathcal{X}_1, x_2 \in \mathcal{X}_2} \|(-Mx_2, x_1^\top M)\|_*$. Telescoping this last bound yields

$$
\sup_n D(x^*, X_n) \leq D(x^*, X_1) + \sum_{k=1}^{\infty} \frac{\gamma_n^2 G^2}{2K} < \infty,
\tag{C.18}
$$

---

[4]This observation was also the starting point of Mertikopoulos et al. [2018] who showed that FTRL in *continuous time* exhibits a similar cycling behavior in zero-sum games with an interior equilibrium.

so $D(x^*, X_n)$ is also bounded from above. Therefore, with $D(x^*, X_n)$ nondecreasing, bounded from above and $D(x^*, X_1) > 0$, it follows that $\lim_{n\to\infty} D(x^*, X_n) > 0$, as claimed. ∎

## D  CONVERGENCE ANALYSIS OF OPTIMISTIC MIRROR DESCENT

We now turn to the *optimistic mirror descent* (OMD) algorithm, as defined by the recursion

$$
\begin{aligned}
X_{n+1/2} &= P_{X_n}(-\gamma_n \hat{g}_n) \\
X_{n+1} &= P_{X_n}(-\gamma_n \hat{g}_{n+1/2})
\end{aligned}
\tag{OMD}
$$

with $X_1$ initialized arbitrarily in dom $\partial h$, and $\hat{g}_n$, $\hat{g}_{n+1/2}$ representing gradient oracle queries at the incumbent and intermediate states $X_n$ and $X_{n+1/2}$ respectively.

The heavy lifting for our analysis is provided by Proposition B.4, which leads to the following crucial lemma:

**Lemma D.1.** *Suppose that* (SP) *is coherent and $g$ is L-Lipschitz continuous. With notation as above and exact gradient input ($\sigma = 0$), we have*

$$
D(p, X_{n+1}) \le D(p, X_n) - \frac{1}{2}\left(K - \frac{\gamma_n^2 L^2}{K}\right)\|X_{n+1/2} - X_n\|^2,
\tag{D.1}
$$

*with $p$ as in Definition 2.1.*

*Proof.* Substituting $x \leftarrow X_n$, $y_1 \leftarrow -\gamma_n g(X_n)$, and $y_2 \leftarrow -\gamma_n g(X_{n+1/2})$ in Proposition B.4, we obtain the estimate:

$$
\begin{aligned}
D(p, X_{n+1}) &\le D(p, X_n) - \gamma_n \langle g(X_{n+1/2}), X_{n+1/2} - p \rangle \\
&\quad + \frac{\gamma_n^2}{2K}\|g(X_{n+1/2}) - g(X_n)\|_*^2 - \frac{K}{2}\|X_{n+1/2} - X_n\|^2 \\
&\le D(p, X_n) + \frac{\gamma_n^2 L^2}{2K}\|X_{n+1/2} - X_n\|^2 - \frac{K}{2}\|X_{n+1/2} - X_n\|^2,
\end{aligned}
\tag{D.2}
$$

where we used the fact that $g$ is $L$-Lipschitz and that $p$ is a solution of (SP) such that (MVI) holds for all $x \in \mathcal{X}$. ∎

We are now finally in a position to prove Theorem 4.1 (reproduced below for convenience):

**Theorem.** *Suppose that* (SP) *is coherent and $g$ is L-Lipschitz continuous. If* (OMD) *is run with exact gradient input and a step-size sequence $\gamma_n$ such that*

$$
0 < \lim_{n\to\infty} \gamma_n \le \sup_n \gamma_n < K/L,
\tag{D.3}
$$

*the sequence $X_n$ converges monotonically to a solution $x^*$ of* (SP), *i.e., $D(x^*, X_n)$ is non-increasing and converges to $0$.*

*Proof.* Let $p$ be a solution of (SP) such that (MVI) holds for all $x \in \mathcal{X}$ (that such a solution exists is a consequence of Definition 2.1). Then, by the stated assumptions for $\gamma_n$, Lemma D.1 yields

$$
D(p, X_{n+1}) \le D(p, X_n) - \frac{1}{2}K(1 - \alpha^2)\|X_{n+1/2} - X_n\|^2,
\tag{D.4}
$$

where $\alpha \in (0, 1)$ is such that $\gamma_n^2 < \alpha K/L$ for all $n$ (that such an $\alpha$ exists is a consequence of the assumption that $\sup_n \gamma_n < K/L$). Now, telescoping (D.1), we obtain

$$
D(p, X_{n+1}) \le D(p, X_1) - \frac{1}{2}\sum_{k=1}^{n}\left(K - \frac{\gamma_k^2 L^2}{K}\right)\|X_{k+1/2} - X_k\|^2,
\tag{D.5}
$$

and hence:

$$
\sum_{k=1}^{n}\left(1 - \frac{\gamma_k^2 L^2}{K^2}\right)\|X_{k+1/2} - X_k\|^2 \le \frac{2}{K}D(p, X_1).
\tag{D.6}
$$

With $\sup_n \gamma_n < K/L$, the above estimate readily yields $\sum_{n=1}^{\infty}\|X_{n+1/2} - X_n\|^2 < \infty$, which in turn implies that $\|X_{n+1/2} - X_n\| \to 0$ as $n \to \infty$.

By the compactness of $\mathcal{X}$, we further infer that $X_n$ admits an accumulation point $x^*$, i.e., there exists a subsequence $n_k$ such that $X_{n_k} \to x^*$ as $k \to \infty$. Since $\|X_{n_k+1/2} - X_{n_k}\| \to 0$, this also implies that $X_{n_k+1/2}$ converges to $x^*$ as $k \to \infty$. Further, by passing to a subsequence if necessary, we may also assume without loss of generality that $\gamma_{n_k}$ converges to some limit value $\gamma > 0$. Then, by the Lipschitz continuity of the prox-mapping (cf. Proposition B.4), we readily obtain

$$x^* = \lim_{k\to\infty} X_{n_k+1/2} = \lim_{k\to\infty} P_{X_{n_k}}(-\gamma_{n_k} g(X_{n_k})) = P_{x^*}(-\gamma g(x^*)), \tag{D.7}$$

i.e., $x^*$ is a solution of (SP).

Since (MVI) holds locally around $x^*$ (by Definition 2.1), the same reasoning as above shows that

$$D(x^*, X_{n+1}) \le D(x^*, X_n) - \frac{1}{2} K(1 - \alpha^2)\|X_{n+1/2} - X_n\|^2 \le D(x^*, X_n), \tag{D.8}$$

for all sufficiently large $n$. This shows that $D(x^*, X_n)$ is non-decreasing; since $\liminf_{n\to\infty} D(x^*, X_n) = 0$ (by Bregman reciprocity), we ultimately conclude that $\lim_{n\to\infty} D(x^*, X_n) = 0$, i.e., $X_n \to x^*$. ∎

Our last result concerns the convergence of (OMD) in strictly coherent problems with a stochastic gradient oracle:

*Proof of Theorem 4.3.* Our argument hinges on the inequality

$$D(x^*, X_{n+1}) \le D(x^*, X_n) - \gamma_n\langle \hat{g}_{n+1/2}, X_{n+1/2} - x^*\rangle + \gamma_n^2/(2K) \|\hat{g}_{n+1/2} - \hat{g}_n\|_*^2 \tag{D.9}$$

which is obtained from the two-point estimate (B.22b) by substituting $x \leftarrow x^*$, $x_1 \leftarrow X_n$, $y_1 \leftarrow \hat{g}_n$, $x_1^+ \leftarrow X_{n+1/2} = P_{X_n}(-\gamma_n\hat{g}_n)$, $y_2 \leftarrow \hat{g}_{n+1/2}$, and $x_2^+ \leftarrow X_n = P_{X_n}(-\gamma_n\hat{g}_{n+1/2})$. Then, working as in the proof of Proposition C.1, we obtain the following estimate for the sequence $D_n = D(x^*, X_n)$:

$$D_{n+1} \le D_n - \gamma_n\langle g(X_{n+1/2}), X_{n+1/2} - x^*\rangle - \gamma_n\langle U_{n+1}^+, X_n - x^*\rangle + \frac{\gamma_n^2}{2K}\|\hat{g}_{n+1/2} - \hat{g}_n\|_*^2$$

$$\le D_n + \gamma_n\xi_{n+1}^+ + \frac{\gamma_n^2}{K}\Big[\|\hat{g}_n\|_*^2 + \|\hat{g}_{n+1/2}^2\|_*\Big], \tag{D.10}$$

where $U_{n+1}^+ = \hat{g}_{n+1/2} - g(X_{n+1/2})$ denotes the martingale part of $\hat{g}_{n+1/2}$ and we have set $\xi_{n+1}^+ = \langle U_{n+1}^+, X_{n+1/2} - x^*\rangle$. Since $\mathbb{E}[\|g_n\|_*^2 \mid X_n, \ldots, X_1]$ and $\mathbb{E}[\|g_{n+1/2}\|_*^2 \mid X_{n+1/2}, \ldots, X_1]$ are both bounded by $G^2$, we get the bound

$$\mathbb{E}[D_{n+1} \mid \mathcal{F}_n] \le D_n + \frac{G}{K}\gamma_n^2. \tag{D.11}$$

Following the same steps as in the proof of Proposition C.1, it then follows that $D_n$ converges to some limit value $D_\infty$.

To proceed, telescoping (D.10) also yields

$$D_{n+1} \le D_1 - \sum_{k=1}^n \gamma_k\langle g(X_{k+1/2}), X_{k+1/2} - x^*\rangle + \sum_{k=1}^n \gamma_k\xi_{k+1}^+ + \sum_{k=1}^n \frac{\gamma_k^2}{2K}\|\hat{g}_{k+1/2} - \hat{g}_k\|_*^2. \tag{D.12}$$

Each term in the above bound can be controlled in the same way as the corresponding terms in (C.4). Thus, repeating the steps in the proof of Proposition C.2, it follows that there exists a subsequence of $X_{n+1/2}$ (and hence also of $X_n$) which converges to $x^*$. Our claim then follows by combining the two intermediate results above in the same way as in the proof of Theorem 3.1(a); to avoid needless repetition, we omit the details. ∎

## E    EXPERIMENTAL RESULTS

### E.1    ADAM WITH EXTRA-GRADIENT STEP

For most of our experiments, the method that seemed to generate the best results was Adam and its optimistic version [Daskalakis et al., 2018]; for a pseudocode iplementation, see Algorithm 3 below. We also noticed empirically that it was more efficient to use two different sets of moment estimates $(m_t, v_t)$ and $(m_t', v_t')$ for the first and the second gradient steps. We used this algorithm for our experiments with both GMMs and the CelebA/CIFAR-10 datasets.

---

**Algorithm 3:** Adam with extra-gradient add-on (optimistic Adam)

---

Compute stochastic gradient: $\nabla_{\theta,t}$

Update biased estimate of 1st momentum: $m_t = \beta_1 m_{t-1} + (1 - \beta_1)\nabla_{\theta,t}$

Update biased estimate of 2nd momentum: $v_t = \beta_2 v_{t-1} + (1 - \beta_2)\nabla_{\theta,t}^2$

Compute bias corrected 1st moment: $\hat{m}_t = \frac{m_t}{1-\beta_1^t}$

Compute bias corrected 2nd moment: $\hat{v}_t = \frac{v_t}{1-\beta_2^t}$

Perform: $\theta_t' = \theta_{t-1} - \eta \frac{\hat{m}_t}{\sqrt{\hat{v}_t}+\epsilon}$

Compute stochastic gradient: $\nabla_{\theta',t}$

Update biased estimate of 1st momentum: $m_t' = \beta_1 m_{t-1}' + (1 - \beta_1)\nabla_{\theta',t}$

Update biased estimate of 2nd momentum: $v_t' = \beta_2 v_{t-1}' + (1 - \beta_1)\nabla_{\theta',t}^2$

Compute bias corrected 1st moment: $\hat{m}_t' = \frac{m_t'}{1-\beta_1^t}$

Compute bias corrected 2nd moment: $\hat{v}_t' = \frac{v_t'}{1-\beta_1^t}$

Perform: $\theta_t = \theta_{t-1} - \eta' \frac{\hat{m}_t'}{\sqrt{\hat{v}_t'}+\epsilon}$

Return $\theta_t$

---

**Table 1:** Generator and discriminator architectures for our images experiments

| Generator |
|---|
| latent space 100 (gaussian noise) |
| dense 4 × 4 × 512 batchnorm ReLU |
| 4×4 conv.T stride=2 256 batchnorm ReLU |
| 4×4 conv.T stride=2 128 batchnorm ReLU |
| 4×4 conv.T stride=2 64 batchnorm ReLU |
| 4×4 conv.T stride=1 3 weightnorm tanh |

| Discriminator |
|---|
| Input Image 32×32×3 |
| 3×3 conv. stride=1 64 lReLU |
| 3×3 conv. stride=2 128 lReLU |
| 3×3 conv. stride=1 128 lReLU |
| 3×3 conv. stride=2 256 lReLU |
| 3×3 conv. stride=1 256 lReLU |
| 3×3 conv. stride=2 512 lReLU |
| 3×3 conv. stride=1 512 lReLU |
| dense 1 |

## E.2 EXPERIMENTS WITH STANDARDS DATASETS

In this section we present the results of our image experiments using OMD training techniques. Inception and FID scores obtained by our model during training were reported in Fig. 3: as can be seen there, the extra-gradient add-on improves the performance of GAN training and efficiently stabilizes the model; without the extra-gradient step, performance tends to drop noticeably after approximately 100$k$ steps.

For ease of comparison, we provide below a collection of samples generated by Adam and optimistic Adam in the CelebA and CIFAR-10 datasets. Especially in the case of CelebA, the generated samples are consistently more representative and faithful to the target data distribution.

### E.2.1 NETWORK ARCHITECTURE AND HYPERPARAMETERS

For the reproducibility of our experiments, we provide Table 1 and Table 2 the network architectures and the hyperparameters of the GANs that we used. The architecture employed is a standard DCGAN architecture with a 5-layer generator with batchnorm, and an 8-layer discriminator. The generated samples were 32×32×3 RGB images.

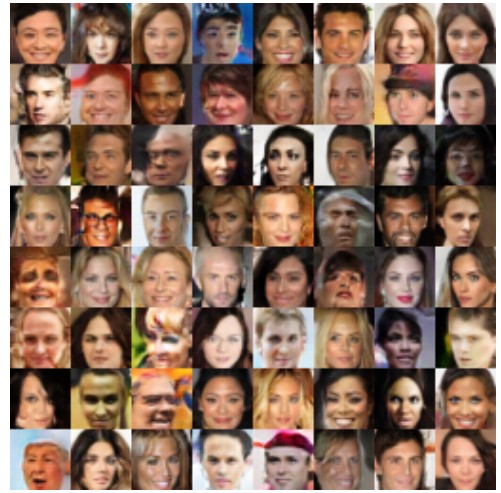 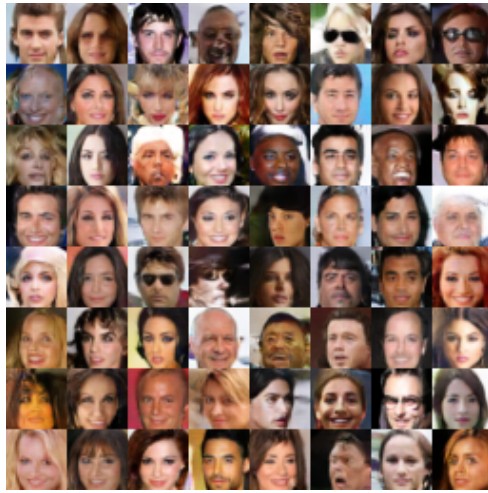

**(a)** Vanilla versus optimistic Adam training in the CelebA dataset (left and right respectively).

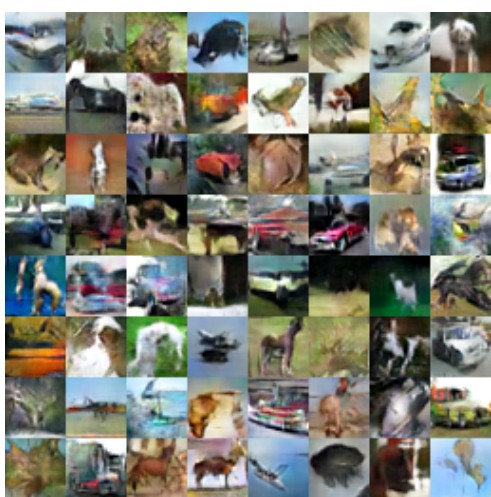 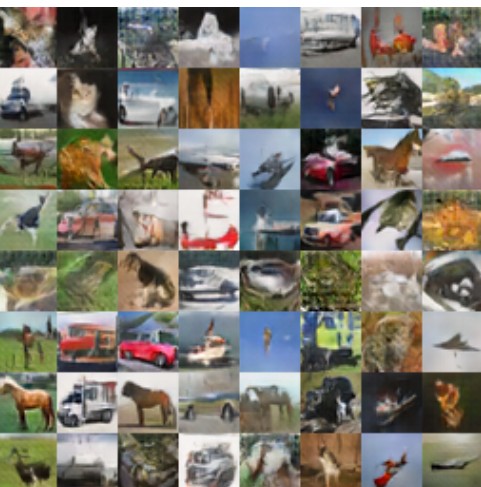

**(b)** Vanilla versus optimistic Adam training in the CIFAR-10 dataset (left and right respectively).

**Figure 6:** GAN training with and without an extra-gradient step in the CelebA and CIFAR-10 datasets.

**Table 2:** Image experiments settings

| |
|---|
| batch size = 64 Adam learning rate = 0.0001 |
| Adam $\beta_1$ = 0.0 |
| Adam $\beta_2$ = 0.9 |
| max iterations = 200000 |
| WGAN-GP $\lambda$ = 1.0 |
| WGAN-GP $n_{dis}$ = 1 |
| GAN objective = 'WGAN-GP' |
| Optimizer = 'extra-Adam' or 'Adam' |

