# OpenReview forum: "Optimistic mirror descent in saddle-point problems: Going the extra (gradient) mile"
_ICLR.cc/2019/Conference_

### Official Review · AnonReviewer2 · 2018-10-31
**Coherent condition is highly related to the pseudo-monotone property in operator theory.**

**Rating:** 5
**Confidence:** 5

**Review:**

Prons:
This paper provides an optimistic mirror descent algorithm to solving minmax optimization problem. Its global convergence is guaranteed under the coherence property. The experimental results are promising.

Cons:
1.	The coherence property is still a strong assumption. The sufficient conditions provided in Corollary 3.2 and 3.3 to guarantee coherence property are too specific to cover existing GAN models.

2.	The current theoretical contribution seems incrementally. From the perspective of operator theory, the coherence property is highly related to the pseudo-monotone property. Extragradient method to solve the pseudo-monotone VIP has already existed in the literature [1]. The proposed OMD can be simply regarded a stochastic extension of [1] and simultaneously generalize the European distance in [1] to Bregman distance.

3.	The integrating of Adam and OMD in the experiments is very interesting. To match the experiments, we highly recommend the authors to show the convergence of OMD + Adam with or without coherence condition, rather than requiring a diminishing learning rate.

[1] Noor, Muhammad Aslam, et al. "Extragradient methods for solving nonconvex variational inequalities." Journal of Computational and Applied Mathematics 235.9 (2011): 3104-3108.

---

> ### Author Response · Authors · 2018-11-14
> **Thanks for the feedback! (see below why pseudo-monotonicity is quite different)**
>
> We thank the reviewer for their constructive remarks! We reply point-by-point below:
>
> 1.	To be sure, coherence does not cover all GAN problems: GANs can be so complex that we feel that any endeavor to account for all problems would be chimeric (at least, given our current level of understanding of the GAN landscape). Being fully aware of this, our goal in this paper was simply to provide concrete theoretical evidence that the inclusion of an extra-gradient step can help resolve many of the problems that arise in practice (and, in particular, cycling and oscillatory mode collapses). In this regard, our paper tackles a significantly wider framework than the 2018 ICLR paper of Daskalakis et al. which only addressed bilinear models.
>
> Furthermore, we would like to point out that Corollaries 3.2 and 3.3 are only *sufficient* conditions for coherence. To make an analogy with convex analysis, in practice, when trying to determine whether a given function is convex, one of the standard techniques is to show that its Hessian matrix is diagonally dominant - and, hence, positive-semidefinite. Obviously, this is just a sufficient condition, but it is still useful in practice. We view Corollaries 3.2 and 3.2 in a similar light: they show that our results cover a wide array of cases of practical (and theoretical) interest, without attempting to be exhaustive.
>
>
> 2.	Regarding the relation with pseudo-monotonicity: despite any formal similarities, we would like to point out that coherence and pseudo-monotonicity can be quite different. As an example, take the objective function (2.2) in our paper: for x_1 = 1/2, we get f(1/2,x_2) = (x_2^2 - 2)^2 (4 + 5x_2^2) / 16, which has *two* well-separated maximizers, i.e. it is not even quasi-concave - implying in turn that (2.2) is not pseudo-monotone (it is, in fact, multi-modal in x_2).
>
> Moreover, as we pointed in our reply to Reviewer 2, the version of coherence that we presented was the simplest possible one (and we did so for reasons of clarity and ease of presentation). Our definition can be weakened substantially by considering the following definition of "weak coherence":
>
> Definition: We say that f is weakly coherent if:
> (i) There exists a solution p of (SP) that satisfies (VI).
> (ii) Every solution x* of (SP) satisfies (VI) locally, i.e., g(x) (x - x*) ≥ 0 for all x sufficiently close to x*.
>
> As we pointed out in our reply to Reviewer 2, under this *weaker* definition of coherence, the solution set of (SP) need no longer be convex, thus making the difference with pseudo-monotone problems even more pronounced. As a very simple example, consider the case where Player 1 controls x,y in [-1,1], and the objective function is f(x,y) = x^2 y^2, i.e., Player 2 has no impact in the game (just for simplicity). In this case, the solution set of the problem is the cross-shaped set X* = {(x,y) : x=0 or y=0}, which is non-convex - in stark contrast to the convex structure of the solution set of pseudo-monotone problems.
>
> We will update our manuscript accordingly as soon as possible to make this change!
>
> We will also include a detailed discussion of the paper by Noor et al. - we were not aware of it, and we thank the reviewer for bringing it to our attention.
>
>
> 3.	Regarding the integration of Adam in our proof technique: we agree with the reviewer that this is a worthwhile extension, but not one that can be properly undertaken without completely changing the structure of the paper and its focus. Adam has a very specific update structure and requires the introduction of significant machinery to handle theoretically, so we do not see how it can be done without greatly shifting the scope and balance of our treatment and analysis.

---

### Official Review · AnonReviewer1 · 2018-11-02
**A first step to handle non-convexity in saddle point optimization**

**Rating:** 6
**Confidence:** 5

**Review:**

This work provides the converge proof of the last iterates of two stochastic methods (almost surely) that the author called  mirror descent and optimistic mirror descent under an assumption weaker than monotonicity called coherence.
Roughly, the definition of coherence is the equivalence between being a  saddle point and the solution of the Minty variational inequality.

Overall, I think that this paper try to tackle an interesting problem which is to prove convergence of saddle point algorithms under weaker assumption than monotonicity of the operator.

However, I have some concerns:

- I think that the properties of coherent saddle point could be more investigated. For instance is the set of coherent saddle point connected ? It would be very relevant for GANs. You claim that "neither strict, nor null coherence imply a unique solution to (SP)," but I do not see any proof of that statement (both provided examples have a unique SP). I agree that you can set $g$ to $0$ in some directions to get an affine space a of saddle points but is there examples where the set of solution is not an affine space (intersected with the constraints) ?
- First of all the results are only asymptotic. (I agree that it can be mitigated saying that there is (almost) no results on non-monotone VI and it is a first step to try to handle non-convexity of the objective functions.)
- One big pro of this work might have been new proof techniques to handle non-monotonicity in variational inequalities but the coherence assumption looks like to be the weakest condition to use the standard proof technique of convergence of the (MD) and (OMD). Nevertheless, this work is still interesting since it handles in a subtle way stochasticity (I did not have time to check Theorem 2.18 [Hall & Heyde 1980], I would be good to repeat it in the appendix for self-completeness)
- This work could be easily extended to non zero-sum games which is crucial in practice since most of the state of the art GANs (such as WGAN with gradient penalty or non saturating GAN) are non zero-sum games.
- Are you sure of the use of the denomination Optimistic mirror descent ? What you are presenting is the extragradient method. These two methods are slightly different, If you look at (5) in (Daskalaki et al., 2018) you'll notice that the updates are slightly different from you (OMD), particularly (OMD) require two gradient computations per iteration whereas (5) in (Daskalaki et al., 2018) requires only one. (it just requires to memorize the previous gradient)

Minor comment:
- For saddle point (and more generally variational inequalities) Mirror descent is no longer a descent algorithm. The name used by the literature is mirror-prox method (see Juditsky's paper)
- in (C.1) U_n is not defined anywhere but I guess it is $\hat g_n - g(X_n)$.
- Some cited paper are published paper but cited as arXiv paper.
- Lemma D.1 could be extended to the case (\sigma \neq 0) but the additional noise term might be hard to handle to get a result similar as Thm 4.1
for $\sigma \neq 0$.

---

> ### Author Response · Authors · 2018-11-14
> **Thanks for the feedback!**
>
> We thank the reviewer for their in-depth remarks and positive evaluation! We reply point-by-point below:
>
>
> 1.	Regarding the structure of the solution set of a coherent problem: we agree that this structural question can be investigated further but, given space constraints, we are concerned that this might potentially dilute the focus of the paper. Nevertheless, we would like to take advantage of the openreview format to answer in detail the referee's questions regarding the solution set of a coherent problem:
> - As the referee already points out, uniqueness can be easily taken care of by considering the constant function: the solution set of this problem is the entire feasible region, though the problem is null coherent [and vacuously strictly coherent if we interpret Definition 2.1 to hold for the empty set in the case of strict coherence.] More interesting examples with a zeroed-out direction also exist: for instance, the problem f(x_1,x_2) = x_1^2 is strictly coherent, but its solution set is an affine space.
> - Whether the solution set is an affine space intersected with the set of constraints: in the current formulation, it can be shown that the solution set of a coherent problem is a convex space, though not necessarily one obtained as the intersection of an affine set with the feasible region. [We can provide a concrete example if the referee finds this useful]
> - However, as we state in the paper, the definition of coherence can be weakened substantially, and our results still go through. Specifically, consider the following definition of "weak coherence":
>
> Definition: We say that f is weakly coherent if:
> (i) There exists a solution p of (SP) that satisfies (VI).
> (ii) Every solution x* of (SP) satisfies (VI) locally, i.e., g(x) (x - x*) ≥ 0 for all x sufficiently close to x*.
>
> Under this *weaker* definition of coherence, the solution set of (SP) need no longer be convex! To see this, consider a very simple optimization example where Player 1 controls x,y in [-1,1], and the objective function is f(x,y) = x^2 y^2 (i.e., Player 2 has no impact in the game, just for simplicity). In this case, the solution set of the problem is the cross-shaped set X* = {(x,y) : x=0 or y=0}, which is non-convex!
>
> We chose to focus on the case where the solutions of (SP) and (VI) coincide for simplicity and clarity of presentation; however, we will update our manuscript accordingly as soon as possible to make this change!
>
>
> 2.	Indeed, the results are only asymptotic - but, as the reviewer states, we know of virtually no other results at this level of generality, and the analysis has to start somewhere. We agree that getting rates is an important problem, but we believe that all this cannot be addressed within a single paper.
>
>
> 3.	Regarding the similarity of proof techniques with MD/OMD: we would like to point out that conventional MD/OMD proof techniques are typically quite different as they focus on the convergence of the so-called "ergodic average" of the sequence of iterates (see e.g., the cited literature by Nemirovski, Nesterov, Juditski et al., and many others). Averaging techniques rely crucially on the problem being convex-concave and cannot be used in a non-monotone setting; as a result, we took a completely different approach relying on a quasi-Fejér analysis inspired by recent work on Bregman proximal methods in operator theory.
>
>
> 4.	We concur that our results can be extended to non-zero-sum games, this is a great observation! Again, we did not make this link explicit in our paper for simplicity, but we will definitely update our manuscript accordingly.
>
>
> 5.	Regarding the name "optimistic mirror descent". In the original NIPS 2013 paper of Rakhlin and Sridharan, the authors present two variants of OMD: one is essentially the mirror-prox algorithm of Nemirovski (2004), and the other is a "momentum"-like variant which was further studied by Daskalakis et al. in their recent 2018 ICLR paper. Regrettably, there is a fair bit of confusion in the literature regarding what "optimistic" descent is: personally, we have a strong preference for the original "mirror-prox" terminology of Nemirovski (after all, in saddle-point problems, the method is *not* a descent method). However, we used the OMD terminology of Rakhlin and Sridharan because it seems to be more easily recognizable in the GAN community.
>
>
> 6. Minor comments: We will take care of those, thanks!

---

> > ### Comment · AnonReviewer1 · 2018-12-04
> > **Thank you for you detailed answer**
> >
> > Thank you for you detailed answer.
> >
> > "[We can provide a concrete example if the referee finds this useful]" would love to.
> >
> > Regarding 3. I would like to say that the strict coherence assumption is an extension of the strict monotonicity assumption with which you can also prove last iterate converge. Nemirovski, Nesterov, Juditski focus on general monotonicity (the equivalent of you general coherence with which you do not prove any last iterate convergence result)
> > An interesting point I would like to make is that Last iterate convergence have been proven in the literature under the *strong* monotonicity assumption see for instance [Chen et al. 1997] (the Forward-backward algorithm is a generalization of the MD algorithm). Maybe you could have convergence rate under a *strong* coherence assumption (but also raising the question to what extend *strong* coherence assumption is realistic)
> >
> >
> > Chen, George HG, and R. Tyrrell Rockafellar. "Convergence rates in forward--backward splitting." SIAM Journal on Optimization 7.2 (1997): 421-444.

---

> > > ### Author Response · Authors · 2018-12-04
> > > **Thanks for the extra round of feedback!**
> > >
> > > Many thanks for the extra round of feedback and the encouraging remarks! We reply to the points you raised below:
> > >
> > > 1. Regarding the example of a coherent problem with a general convex solution set.
> > >
> > > Again, for simplicity, focus on the optimization case, i.e., the minimization of a function f:X->R (X convex). In this case, letting X* = argmin f, and writing g(x) for the (sub)gradient of f, the (strict) coherence requirement takes the form:
> > >    - <g(x),x-x*>≥0 for all x in X and all x* in X*.
> > >   - Equality holds above if and only if x lies in X*.
> > >
> > > Now, fix some convex subset C of X, and let f(x) = dist(x,C)^2 (where dist denotes the standard Euclidean setwise distance). By construction, f is convex (though not strictly so) and X*=C. Convexity guarantees the first requirement of coherence. For the second, note that g(x) is a multiple of x - proj_C(x) so, for any x* in X*, the product <g(x),x-x*> vanishes only if x lies itself in C (since C=X*).
> > >
> > > Of course, the above function is convex, but if we perturb f away from C = X* in an appropriate way, non-convex examples can also be constructed (though there are diminishing returns regarding the simplicity of the resulting example).
> > >
> > > [NB: just to avoid any misunderstanding, the above concerns the definition of coherence as presented in the *original* version of the paper; the current version includes examples with non-convex solution sets like x^2 y^2 as we outlined in our first reply.]
> > >
> > >
> > > 2. Thanks for the pointer to Chen and Rockafellar, it looks very promising for future study! The reviewer's suggestion seems very plausible but the devil is often in the details, so we would need more time in order to provide a more definitive reply.
> > >
> > >
> > > We cannot revise the paper at this time, but we'd of course be happy to do so along the lines above if accepted.

---

### Official Review · AnonReviewer3 · 2018-11-04
**A good paper**

**Rating:** 7
**Confidence:** 3

**Review:**

This paper is trying to find a saddle-point of a Lagrangian using mirror descent. Mirror descent based methods use Bregman divergence to encode the convexity and smoothness of objective function beyond the euclidean structure. The main contribution of this paper is adding an extra gradient step to the standard MD, i.e., step 5 in Algorithm 2 as well as stochastic versions. Numerical experiments support their results.

---

> ### Author Response · Authors · 2018-11-14
> **Thanks for the feedback!**
>
> We thank the reviewer for their positive and encouraging feedback! We also feel that the inclusion of an extra-gradient step can greatly enhance the stability of GAN training methods, and can provide further key insights.

---

### Public Comment · (anonymous) · 2019-02-06
**Proof of Theorem 4.1**

Hi, very nice paper! In the proof of Theorem 4.1 (in appendix D, end of page 19), where you only have coherence (not necessarily strict), it seems like you only establish convergence of OMD to some point but not necessarily to x^*. Am I missing something?

---

> ### Author Response · Authors · 2019-02-06
> **Convergence to some solution, yes, not necessarily the x* in the proof, no**
>
> Hey,
>
> Thanks for your comments and positive feedback!
>
> Yes, in the proof of Theorem 4.1, x* is a "special" solution point which satisfies the variational inequality formulation (VI) of the saddle-point problem globally. This point is used to establish convergence to the solution set of the problem, but it is not necessarily the end state of the algorithm - i.e., it is not the "solution point x*" alluded to in the statement of the theorem.
>
> Thanks for catching this ambiguity, typo correction on its way!

---

### Meta-Review · Area_Chair1 · 2018-12-18
**Some progress for analysis of non-monotone variational inequalities**

**Confidence:** 4
**Recommendation:** Accept (Poster)

**Metareview:**

This paper investigates the usage of the extragradient step for solving saddle-point problems with non-monotone stochastic variational inequalities, motivated by GANs. The authors propose an assumption weaker/diffrerent than the pseudo-monotonicity of the variational inequality for their convergence analysis (that they call "coherence"). Interestingly, they are able to show the (asympotic) last iterate convergence for the extragradient algorithm in this case (in contrast to standard results which normally requires averaging of the iterates for the stochastic *and* mototone variational inequality such as the cited work by Gidel et al.). The authors also describe an interesting difference between the gradient method without the extragradient step (mirror descent) vs. with (that they called optimistic mirror descent).

R2 thought the coherence condition was too related to the notion of pseudo-monoticity for which one could easily extend previous known convergence results for stochastic variational inequality. The AC thinks that this point was well answered by the authors rebuttal and in their revision: the conditions are sufficiently different, and while there is still much to do to analyze non variational inequalities or having realistic assumptions, this paper makes some non-trivial and interesting steps in this direction. The AC thus sides with expert reviewer R1 and recommends acceptance.